# COVID-19 immune signatures in Uganda persist in HIV co-infection and diverge by pandemic phase

Little is known about the pathobiology of SARS-CoV-2 infection in sub-Saharan Africa, where severe COVID-19 fatality rates are among the highest in the world and the immunological landscape is unique. In a prospective cohort study of 306 adults encompassing the entire clinical spectrum of SARS-CoV-2 infection in Uganda, we profile the peripheral blood proteome and transcriptome to characterize the immunopathology of COVID-19 across multiple phases of the pandemic. Beyond the prognostic importance of myeloid cell-driven immune activation and lymphopenia, we show that multifaceted impairment of host protein synthesis and redox imbalance define core biological signatures of severe COVID-19, with central roles for IL-7, IL-15, and lymphotoxin-α in COVID-19 respiratory failure. While prognostic signatures are generally consistent in SARS-CoV-2/HIV-coinfection, type I interferon responses uniquely scale with COVID-19 severity in persons living with HIV. Throughout the pandemic, COVID-19 severity peaked during phases dominated by A.23/A.23.1 and Delta B.1.617.2/AY variants. Independent of clinical severity, Delta phase COVID-19 is distinguished by exaggerated pro-inflammatory myeloid cell and inflammasome activation, NK and CD8⁺ T cell depletion, and impaired host protein synthesis. Combining these analyses with a contemporary Ugandan cohort of adults hospitalized with influenza and other severe acute respiratory infections, we show that activation of epidermal and platelet-derived growth factor pathways are distinct features of COVID-19, deepening translational understanding of mechanisms potentially underlying SARS-CoV-2-associated pulmonary fibrosis. Collectively, our findings provide biological rationale for use of broad and targeted immunotherapies for severe COVID-19 in sub-Saharan Africa, illustrate the relevance of local viral and host factors to SARS-CoV-2 immunopathology, and highlight underemphasized yet therapeutically exploitable immune pathways driving COVID-19 severity.

The COVID-19 pandemic is the greatest global health crisis in over a century[1]. In high-income countries (HICs), clinical outcomes for patients with severe COVID-19 have improved substantially over time, in part due to targeted administration of immunomodulatory therapeutics (i.e., corticosteroids, interleukin-6 and JAK1/2 antagonists) and widespread SARS-CoV-2 vaccine uptake[2–7]. Effective use of immunomodulatory agents was driven by a multitude of basic and translational investigations in HICs that identified dysregulated pro-

✉ e-mail: mjc2244@columbia.edu

inflammatory immune responses, particularly myeloid cell-driven induction of innate immune signaling and high-levels of inflammatory cytokines and chemokines, as key pathobiological features in severe COVID-19[8]. Despite this, host responses across the clinical spectrum of SARS-CoV-2 infection are heterogeneous, and interplay between inflammatory, metabolic, and microvascular pathways in COVID-19 immunopathology remains incompletely understood[9,10]. The contribution of SARS-CoV-2 variants and HIV co-infection to COVID-19 pathobiology, factors which vary substantially across geographic and income settings worldwide, is also poorly defined.

In sub-Saharan Africa, a low-income region where SARS-CoV-2 vaccine coverage remains poor and critical care capacity is limited, fatality rates for severe COVID-19 are among the highest in the world[11–15]. Although the immunological landscape of the region is unique due to young age demographics, high HIV burden, and distinctive SARS-CoV-2 variants, mechanisms that mediate immunopathology and drive COVID-19 severity in sub-Saharan Africa remain largely unknown[16]. In this context, comprehensive immune profiling across the severity spectrum of COVID-19 is essential to reinforce biological rationale for therapeutic immunomodulation, determine the importance of locally relevant viral and host factors, and identify prognostically-important molecular signatures that may represent potential treatment targets.

In a nationally-representative prospective cohort study in Uganda, we apply multimodal methods to dissect the immunopathology of COVID-19 across three variant-driven phases of the pandemic. We show that while prognostic immune signatures are generally consistent in SARS-CoV-2/HIV co-infection, Delta phase COVID-19 is defined by exaggerated pro-inflammatory myeloid cell and inflammasome activation and NK and CD8+ T cell depletion. Combining these analyses with a contemporary Ugandan cohort of adults hospitalized with influenza and other severe respiratory infections, we show that activation of epidermal and platelet-derived growth factor pathways are distinguishing features of COVID-19, deepening translational understanding of mechanisms potentially underlying pulmonary fibrosis and other clinical sequelae of SARS-CoV-2 infection.

## Results
### Study site, capacity, and patient population
We conducted a prospective observational cohort study (Research in the Epidemiology of Severe and Emerging Infections in Uganda—Coronavirus disease 2019; RESERVE-U-C19) of patients with laboratory (PCR)-confirmed SARS-CoV-2 infection admitted to Entebbe Regional Referral Hospital (ERRH), a 200-bed public hospital in central Uganda. During the study period, ERRH functioned as a national referral hospital for COVID-19; patients with SARS-CoV-2 infection nationwide were referred to the facility for management. Details on study site capacity and available respiratory support are provided in the Methods section.

As part of RESERVE-U-C19, we prospectively obtained clinical data, serum, and whole-blood RNA samples from adults (age ≥ 18 years) with laboratory-confirmed SARS-CoV-2 infection admitted to ERRH from March 22nd, 2020 to July 14th, 2021 (Supplementary Data 1, Supplementary Fig. 1). Both males and females were enrolled; there was no preference for enrollment by self-reported sex. At the conclusion of enrollment, <5% of the Ugandan population was fully vaccinated against SARS-CoV-2[11]. Among enrolled participants (N = 306), most were young adult males and 11% (33/302) were living with HIV (PLWH), of whom 83% (24/29) had suppressed viral loads. Malaria or tuberculosis co-infection was rare (Supplementary Data 1).

Based on World Health Organization (WHO) criteria, we stratified patients into four groups of clinical severity: asymptomatic (N = 66 [21.6%]), mild (N = 149 [48.7%]), moderate (N = 21 [6.9%]), and severe (N = 70 [22.9%]) (Supplementary Data 1; see Supplemental Methods for specific criteria)[17]. Among patients with severe COVID-19, of whom 90.0% (63/70) received oxygen therapy and 88.6% (62/70) received corticosteroids, 17.1% (12/70) died in hospital or were transferred to Uganda's highest level public referral hospital (where more advanced respiratory support was available) due to progressive illness severity.

### SARS-CoV-2 variant-driven pandemic phases
Using national genomic and epidemiologic surveillance data generated by the Uganda Virus Research Institute (UVRI) and WHO, we divided our study period into three pandemic phases defined by dominant circulation of different SARS-CoV-2 variants (various A/B lineages, A.23/A.23.1, Delta) and trends in national SARS-CoV-2 case counts (Supplementary Data 1, Fig. 1a)[18–20]. While patients of varying clinical severity were enrolled during each pandemic phase, a high proportion of patients hospitalized during the Delta variant-driven phase, during which time up to 90% of nationally sequenced samples were the Delta B.1.617.2 and AY sub-lineages, had severe COVID-19[18].

### Increased COVID-19 severity in Uganda is dominated by mediators of innate myeloid cell and Th1/Th17 pathways that persist in PLWH
To determine the relationship between COVID-19 severity and key domains of the host response to viral respiratory infection, we performed targeted profiling of the peripheral blood proteome, measuring 48 soluble immune mediators in serum using a multiplexed immunoassay (N = 306). Across the clinical spectrum, patients with more severe illness exhibited a pro-inflammatory immune signature dominated by mediators of monocyte/macrophage and NK cell activation/chemotaxis and Th1/Th17 pathways (e.g., IL-6, IL-7, IL-15, IL-17F, CXCL9, CXCL10, lymphotoxin-α, TNF). In parallel, patients with severe COVID-19 showed reduced concentrations of key Th2-related mediators (MDC, IL-5) and likely compensatory production of the anti-inflammatory mediator IL-10 (Fig. 1b, c, Supplementary Table 1). More severe COVID-19 was also associated with higher concentrations of key pro-fibrotic mediators of epithelial regeneration/repair (TGF-α), and endothelial activation/permeability (VEGF-A) (Fig. 1b, c). No significant differences in IFN-α2 or IFN-γ concentrations were observed across severity groups (Supplementary Table 1). Findings were consistent in Benjamini-Hochberg [BH]-adjusted multivariable proportional odds models including age, self-reported sex, HIV co-infection, SARS-CoV-2 variant phase, and pre-enrollment symptom duration, the latter included when asymptomatic individuals were excluded (Supplementary Tables 2, 3).

Given the considerable number of mediators associated with COVID-19 severity, we determined the importance of each in distinguishing patients with severe COVID-19 using a gradient-boosted machine classifier model and Shapley additive explanations (SHAP) values. This classifier reinforced the prognostic importance of monocyte/macrophage, NK cell, and Th1-pathway activation, with prediction of severe COVID-19 driven by higher concentrations of IL-7, CXCL10, IL-15, and CXCL9, and lower concentrations of MDC and IL-5 (Fig. 1d, e). TGF-α, a key mediator of the EGFR pathway, was also a high-ranking driver of model prediction, suggesting that concurrent activation of pro-fibrotic epidermal growth factors are important features of severe COVID-19.

To determine if severity-related host responses were consistent in PLWH, we repeated our analyses using two approaches: restricting between-group comparisons to PLWH and including an interaction term between HIV status and each mediator concentration in multivariable proportional odds models of COVID-19 severity. Although limited by the relatively small proportion of PLWH enrolled in our study, analyses using both methods were consistent with those in the larger cohort, suggesting that prognostic host responses in COVID-19 are generally conserved in PLWH (Supplementary Table 4 and Supplementary Fig. 2). While further limited by the small number of PLWH with severe illness, interaction models also suggested that higher concentrations of IFN-α2, IL-6, CCL3, IL-1a, and IL-1Ra were associated with more severe COVID-19 in PLWH (Fig. 1f and Supplementary Fig. 3).

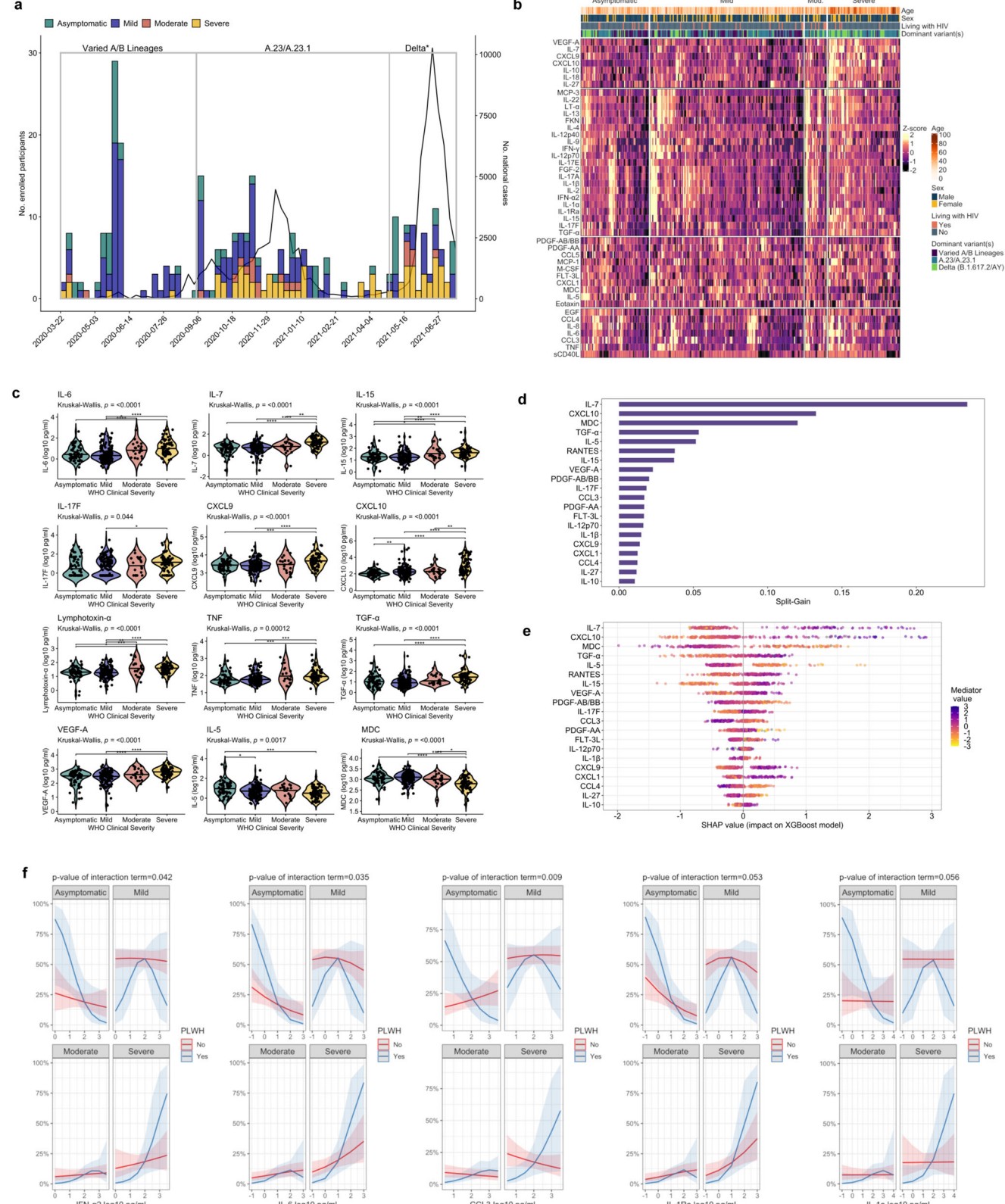

## Whole-blood transcriptional profiling reinforces pro-inflammatory myeloid cell-driven immune activation and variable lymphocyte signatures in severe COVID-19

Of the 306 adults prospectively enrolled in RESERVE-U-C19, RNAseq was performed for a subset of 100 who had whole-blood RNA samples collected simultaneously with serum during each phase of the pandemic. These 100 patients spanned the COVID-19 severity spectrum with demographics, HIV and Delta phase prevalence, and WHO severity classifications comparable to the larger cohort (Supplementary Data 2). Gene set enrichment analysis (GSEA), structured on differential gene expression between patients with and without severe COVID-19 (adjusted for age, self-reported sex, HIV co-infection, and SARS-CoV-2 variant phase), was used to infer biological pathway enrichment associated with COVID-19 severity. At a false discovery rate

**Fig. 1 | COVID-19 pandemic phases and relationships between soluble immune mediators and COVID-19 severity. a** Epidemic curve of study enrollment period; enrolled patients ($N = 306$) assigned to epidemic weeks based on date of hospitalization and colored according to World Health Organization (WHO) clinical severity classification. Right-sided y-axis reflects national SARS-CoV-2 case counts as per the WHO COVID-19 Uganda dashboard. Pandemic phases were defined based on dominant circulation of different variants and trends in national SARS-CoV-2 case counts (Varied A/B lineage phase, $N = 97$; A.23/A.23.1 phase, $N = 141$; Delta phase, $N = 68$). *Delta variant includes B.1.617.2 and AY.1, AY.4, AY.33, AY.39, AY.46, AY.46.4 sublineages. **b** Heatmap of 45 soluble immune mediator concentrations (values $\log_{10}$-transformed, centered, and scaled) stratified by WHO clinical severity classification with rows split by $k$-means clustering ($N = 306$). Individual patient columns are ordered based on differences in mean $z$-score and annotated with age, self-reported sex, HIV status, and dominant SARS-CoV-2 variant at time of hospitalization. IL-3, GM-CSF, and G-CSF omitted from heatmap given large proportion of values below the lower limit of quantification. **c** Soluble mediator concentrations stratified by WHO clinical severity classification ($N = 306$). Concentrations across groups compared using Kruskal-Wallis H test followed by Dunn's test for multiple comparisons with $p$ values adjusted using Benjamini-Hochberg method; *$p < 0.05$, **$p < 0.01$, ***$p < 0.001$, ****$p < 0.0001$. **d** Importance of immune mediators in gradient-boosted machine classifier for prediction of severe ($N = 70$) vs. non-severe (asymptomatic, mild, or moderate, $N = 236$) COVID-19 as per model split-gain values; top 20 variables presented in descending order of importance. **e** Shapley additive explanations (SHAP) values derived from gradient-boosted machine model for prediction of severe ($N = 70$) vs. non-severe ($N = 236$) COVID-19; SHAP values $> 0$ indicate positive impact on prediction while values $< 0$ indicates negative impact (e.g., high concentrations of IL-7 have a strong positive contribution to prediction of severe COVID-19). **f** Probabilities of COVID-19 severity stratified by HIV status ($N = 302$; 4 patients without definitive assessment of HIV status excluded); shading indicates 95% confidence intervals; probabilities derived from multivariable proportional odds model including WHO clinical severity as ordinal dependent variable and age, self-reported sex, and interaction term between HIV status and $\log_{10}$-mediator concentration as independent variables; $p$ values reflect two-sided Wald test of interaction term; PLWH persons living with HIV. Source data are provided in the Source Data file.

(FDR) $q$ value threshold of 0.10, GSEA reinforced a severe COVID-19 immune signature defined by broad activation of key pro-inflammatory innate immune pathways (e.g., TLR4, NF-κB, STAT, complement) and cell populations (neutrophils, macrophages, dendritic cells), in addition to increased production of IL-6, IL-8, and TNF superfamily mediators (Fig. 2a). Patients with severe COVID-19 showed evidence of dynamic changes to the lymphocyte compartment. This included concomitant upregulation of Th1 cell differentiation pathways and T cell activation along with reduced quantities of CD8$^+$ T cells and NK cells, as inferred from digital cytometry deconvolution (Fig. 2a, Supplementary Table 5, Supplementary Fig. 4).

### Oxidative stress, endothelial glycocalyx regeneration, and impaired host protein synthesis define the metabolic signature of severe COVID-19

In patients with severe COVID-19, GSEA identified a metabolic profile dominated by upregulated macromolecule catabolism, phagocytic respiratory burst, and production of reactive oxygen species (ROS) (Fig. 2b). While integral to antimicrobial defense, exaggerated production of these species and resulting redox imbalance may induce changes to endothelial and epithelial barrier integrity, propagating recruitment of pro-inflammatory innate cells and amplifying pulmonary and extrapulmonary tissue injury during severe viral respiratory infection[21,22]. Supporting prognostic interplay between pro-inflammatory immune, microvascular, and thrombotic pathway activation, patients with severe COVID-19 showed consistent evidence of endothelial activation and regeneration of key glycocalyx-anchoring proteoglycans (heparan and chondroitin sulfates), with concomitant upregulation of coagulation pathways and platelet aggregation and activation (Fig. 2b).

In vitro data from infected human alveolar epithelial cells suggest that SARS-CoV-2 and other coronaviruses subvert host protein synthesis to enhance viral replication and impede host antiviral defense[23]. Although derived from peripheral blood RNAseq, GSEA showed consistent evidence of impaired transcription and translation in patients with severe COVID-19, including downregulation of RNA polymerase activity, RNA methylation, ribosome biogenesis, and translational initiation (Fig. 2c). Patients with severe COVID-19 also showed upregulation of pathways involved in viral cell entry and trafficking, including proteolysis, endocytosis and phagocytosis, autophagosome organization and lysosome pH maintenance[24].

### Neutrophil, plasma cell, and activated CD4$^+$ and CD8$^+$ T cell abundance correlate with key indicators of COVID-19 respiratory failure

In parallel with GSEA, we inferred the abundance of key immune cell populations using digital cytometry deconvolution and determined relationships between immune cell populations, soluble mediator concentrations, and COVID-19 severity. Principal component analysis (PCA) applied to immune cell subsets revealed separation of patients with severe COVID-19 across the first principal component, with neutrophils, NK cells, and activated CD4$^+$ and CD8$^+$ T cells identified as the main drivers of immune cell variance (Fig. 2d). Hierarchical correlation analysis restricted to symptomatic patients suggested positive relationships between neutrophil, monocyte, plasma cell, and activated CD4$^+$ T cell abundance, indicators of COVID-19 respiratory failure (oxygen saturation, respiratory rate), and severe morbidity (lower Karnofsky Performance Status [KPS]) (Fig. 2e). Conversely, higher CD8$^+$ T cell abundance correlated with higher KPS. Further reinforcing the relevance of pro-inflammatory myeloid cell activation and CD4$^+$/ CD8$^+$ T cell dynamics in severe COVID-19, we observed relationships between these cell types and soluble mediators centrally implicated in COVID-19 severity (Fig. 2f). This included correlations between neutrophils and IL-6, IL-8, MDC, TGF-α, and VEGF-α, between activated CD4$^+$ cells and lymphotoxin-α, IL-6, and IL-7, and between CD8$^+$ T cells and IL-1Ra, IL-6, and MDC.

### Differential innate mononuclear cell-, Th1, and profibrotic host responses are apparent early and throughout the course of severe COVID-19

Considering that patients were enrolled at various time points following symptom onset, we leveraged our cohort to explore changes in soluble mediators over the reported course of illness. For many of the most important mediators identified in predictive models of COVID-19 severity, concentrations appeared divergent early and throughout the course of symptoms, including IL-7, IL-10, lymphotoxin-α, TGF-α, VEGF-A, and MDC (Fig. 3a).

As most symptomatic patients were enrolled after a week of illness (median 9 days), we determined if severity-stratified mediator concentrations differed in patients who presented earlier in their course. Among patients enrolled within 7 days of symptom onset, evidence of pro-inflammatory monocyte/macrophage (IL-1Ra, IL-1β, IL-6, CXCL10), NK cell (IL-15), Th1/Th17-pathway (IL-7, IL-15, IL-17F), and endothelial (VEGF-A) activation were apparent in patients with severe COVID-19 (Fig. 3b, Supplementary Table 6). Moreover, patients with severe COVID-19 had higher concentrations of EGF and TGF-α, suggesting concurrent activation of pro-fibrotic epidermal growth factor pathways within the first week of illness.

### Multi-pathway immune profiles diverge across SARS-CoV-2 variant-driven pandemic phases

Although in vitro and computational experiments suggest that SARS-CoV-2 variants may differentially induce innate immune signaling and pro-inflammatory cytokine production, little remains known about these relationships in vivo[25,26]. Across three study phases distinguished

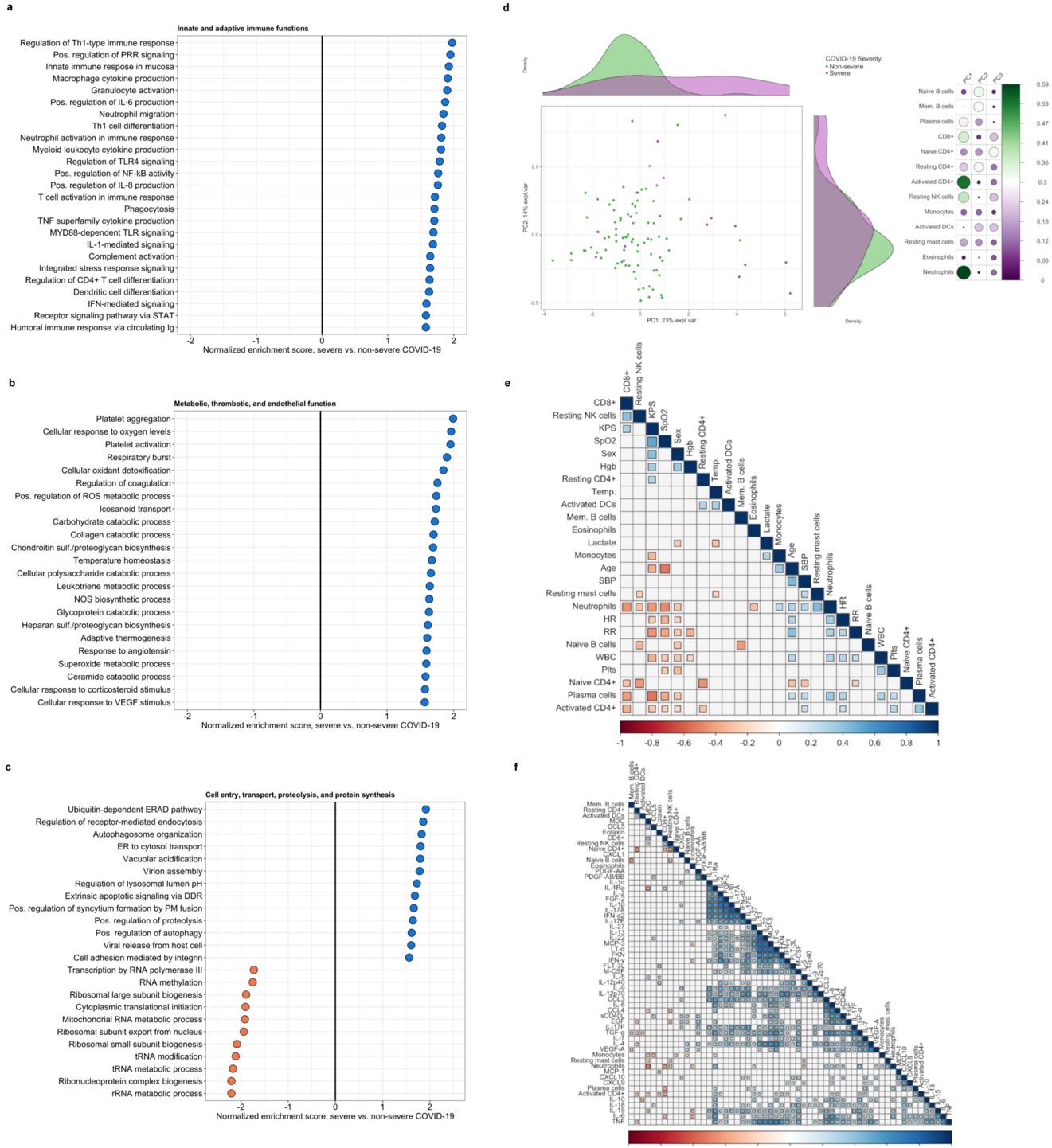

**Fig. 2 | Biological pathway enrichment and immune cell profiles associated with COVID-19 severity. a–c** Differential enrichment of key biological pathways among patients with severe vs. non-severe COVID-19 at false-discovery-rate (FDR) $q$ value ≤ 0.10 ($N = 100$); pathway enrichment determined using Gene Set Enrichment Analysis applied to differentially expressed gene sets generated in a DESeq2 model of whole-blood RNAseq data adjusted for age, self-reported sex, HIV co-infection, and SARS-CoV-2 variant phase (2 patients not known to be living with HIV but with missing rapid diagnostic tests analyzed as HIV negative); PRR pathogen recognition receptor, ROS reactive oxygen species, NOS nitric oxide synthetase, ERAD Endoplasmic-reticulum-associated protein degradation, ER endoplasmic reticulum, DDR death domain receptor. **d** Principal components analysis of immune cell populations in individual patients stratified by COVID-19 severity ($N = 100$); absolute abundance of immune cell populations (naïve and memory B cells, plasma cells, CD8+ T cells, naïve, resting, and activated memory CD4+ T cells,

resting natural killer cells, monocytes, activated dendritic cells, resting mast cells, eosinophils, and neutrophils) inferred from whole-blood RNAseq data using LM22 signature matrix in CIBERSORTx platform; side panel displays squared factor loadings for each immune cell type across the first two principal components; higher loading value indicates greater importance for each cell type in explaining variance across each principal component. **e–f** Hierarchical correlation matrices showing relationships between immune cell abundance, demographics, clinical variables, and immune mediators in patients with symptomatic COVID-19 ($N = 84$); shaded squares reflect Spearman correlation coefficients with a two-sided Benjamini-Hochberg adjusted $p$ value ≤ 0.05; NK natural killer, KPS Karnofsky Performance Status, SpO2 peripheral oxygen saturation, Hgb hemoglobin, Temp temperature, DC dendritic cells, Mem Memory, SBP systolic blood pressure, HR heart rate, RR respiratory rate, WBC white blood cell count, Plts platelet count. Source data are provided in the Source Data file.

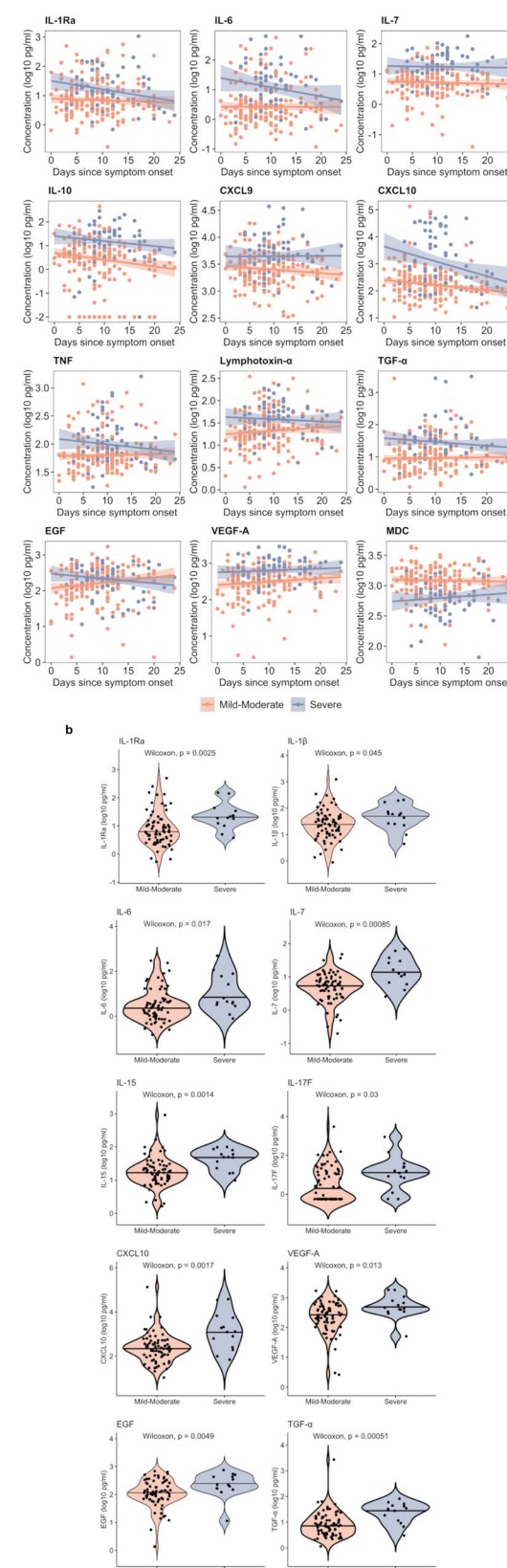

**Fig. 3 | Immune mediator concentrations over the course of symptomatic COVID-19. a** Immune mediator concentrations over the course of COVID-19 symptoms, with robust regression lines and shaded 95% confidence intervals, stratified by COVID-19 severity ($N = 237$; 3 symptomatic patients with extreme outliers in reported illness duration excluded). As an example, an individual data point corresponding to day 0 represents the mediator concentration for a patient whose sample was collected on the day of illness onset, while that corresponding to day 5 represents a patient whose sample was collected on day 5 of illness. **b** Violin plots showing immune mediator concentrations among patients enrolled within 7 days of illness onset stratified by COVID-19 severity ($N = 85$). *P* values reflect two-sided Wilcoxon rank sum tests unadjusted for multiple comparisons; Benjamini-Hochberg-adjusted *P* values are included in Supplementary Table 6. Source data are provided in the Source Data file.

cell differentiation, activation, and chemotaxis (IL-6, IL-8, IL-15, MCP-3, G-CSF, M-CSF, CXCL9, CCL3, CCL4, lymphotoxin-α, TNF) (Fig. 4a, Supplementary Table 7). We also observed evidence of enhanced Th1/Th17-predominant responses (IL-7, IL-17F) and inflammasome activation (IL-18), along with higher concentrations of mediators of profibrotic epithelial regeneration/repair (FGF-2, EGF, TGF-α) and endothelial activation/permeability (VEGF-A) in patients hospitalized during the A.23/A.23.1 and Delta phases (Fig. 4a, Supplementary Table 7).

As concentrations of most mediators peaked in patients with Delta phase COVID-19, we determined the importance of each in distinguishing patients hospitalized during this period using a gradient-boosted machine classifier model and SHAP metrics. This model reinforced the importance of myeloid, NK cell, Th1, and platelet activation in classifying patients with Delta phase COVID-19, with hospitalization during this period driven by higher concentrations of sCD40L, IL-6, IL-15, IL-18, CCL4, and CCL3, as well as compensatory anti-inflammatory, Th2-related mediators (IL-10, IL-1Ra, IL-5), and those reflective of pro-fibrotic epithelial regeneration/repair (TGF-α) (Fig. 4b, c).

To further assess the relationship between host response features and Delta phase COVID-19, we compared mediator concentrations and inferred immune cell abundance between patients admitted during and prior to this period using multivariable linear regression models adjusted for age, self-reported sex, HIV status, corticosteroid exposure, and WHO severity classification. Suggesting that differences in the Delta-phase immune profile are independent of these factors, multivariable models were generally consistent with between-group comparisons reported above (Fig. 4d, e and Supplementary Tables 8, 9). This included significantly higher concentrations of pro-inflammatory innate myeloid, NK cell, and Th1-pathway mediators and profibrotic growth factors. Immune cell models were similarly consistent, with higher neutrophil and lower NK cell and resting CD4⁺ and CD8⁺ T cell abundance in Delta phase COVID-19.

## Whole-blood transcriptional profiling identifies inflammasome assembly as a predominant feature of Delta phase COVID-19

GSEA, structured on differential gene expression between patients with and without Delta phase COVID-19 and co-variable adjusted as above, reinforced an immune profile dominated by pro-inflammatory myeloid cell activation, ROS generation, and impairment of host protein synthesis in patients with Delta phase COVID-19 (Fig. 4f). While activation of many of these processes was observed in patients with severe COVID-19, inflammasome assembly was highly enriched in patients with Delta phase COVID-19, with genes encoding multiple pattern recognition receptors (*NLRP6, NLRP1, TLR6, TLR4, AIM2, TREM2*), Pyrin (*MEFV*), caspase recruitment domain proteins (*CARD8*), cytoplasmic stress granules (*DDX3X*), and phospholipase C (*PLCG2*) comprising the core pathway enrichment set. This suggests that inflammasome activation and pyroptosis may play a key role in amplifying pro-inflammatory responses following host detection of Delta-variant SARS-CoV-2 and related danger

by dominant circulation of different SARS-CoV-2 variants, distinct clinical and host response profiles were apparent, with more severe illness and higher concentrations of most mediators observed during periods dominated by A.23/A.23.1 and Delta B.1.617.2/AY variants (Supplementary Data 3, Fig. 4a, Supplementary Table 7). This included higher concentrations of multiple mediators closely associated with severe COVID-19 and reflective of pro-inflammatory myeloid and NK

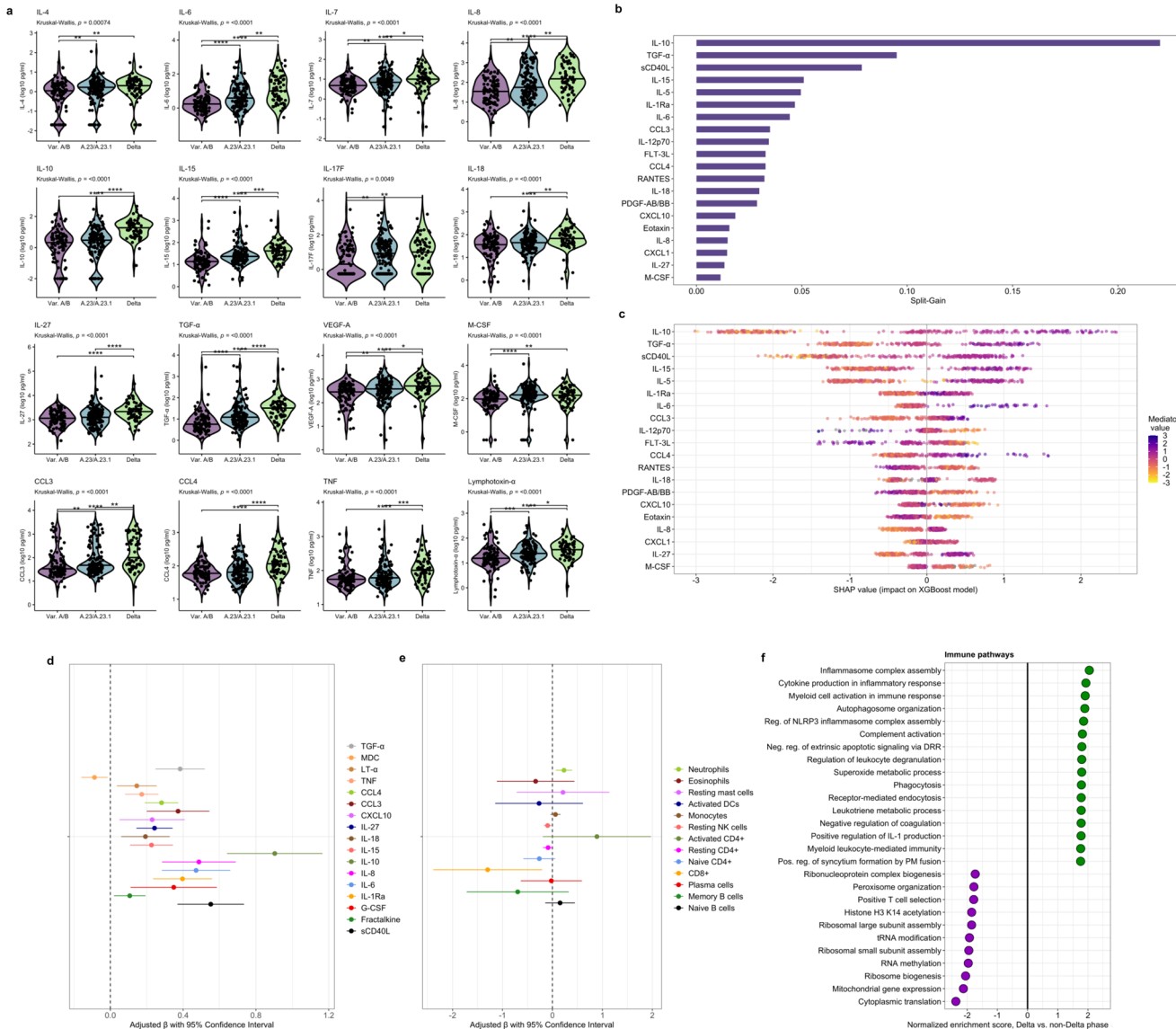

**Fig. 4 | Relationships between immune mediators, cell populations, biological pathways, and COVID-19 pandemic phases. a** Violin plots showing immune mediator concentrations stratified by SARS-CoV-2 variant-driven pandemic phases (variable A/B lineage, A.23/A.23.1, Delta) at time of hospitalization ($N = 306$). Concentrations across groups compared using Kruskal-Wallis H test followed by Dunn's test for multiple comparisons with $p$ values adjusted using Benjamini-Hochberg method; *$p < 0.05$, **$p < 0.01$, ***$p < 0.001$, ****$p < 0.0001$. **b** Importance of immune mediators in gradient-boosted machine classifier for prediction of Delta ($N = 68$) vs. non-Delta phase ($N = 238$; includes variable A/B lineage and A.23/A.23.1 phases) COVID-19 as per model split-gain values; top 20 variables presented in descending order of importance. **c** Shapley additive explanations (SHAP) values derived from gradient-boosted machine model for prediction of Delta ($N = 68$) vs. non-Delta phase ($N = 238$; includes variable A/B lineage and A.23/A.23.1 phases) COVID-19; SHAP values $> 0$ indicate positive impact on prediction while values $< 0$ indicates negative impact (e.g., high concentrations of IL-10 have a strong positive contribution to prediction of Delta phase COVID-19). **d, e** Associations between Delta phase COVID-19 and $log_{10}$-immune mediator concentrations ($N = 306$) and $log_{10}$-

immune cell abundance ($N = 100$); coefficients with 95% confidence interval bars generated in multivariable logistic regression models including hospitalization during Delta phase as a binary dependent variable (vs. hospitalization during variable A/B lineage and A.23/A.23.1 phases) and age (continuous), self-reported sex (binary), HIV co-infection (binary; 4 patients not known to be living with HIV but with missing rapid diagnostic tests analyzed as HIV negative), exposure to corticosteroids (binary), and WHO clinical severity classification (categorical) as independent variables; NK natural killer, DC dendritic cells. **f** Differential enrichment of key biological pathways among patients with Delta vs. non-Delta phase COVID-19 at false-discovery-rate (FDR) $q$ value $\leq 0.10$ ($N = 100$); pathway enrichment determined using Gene Set Enrichment Analysis applied to differentially expressed gene sets generated in a DESeq2 model of whole-blood RNAseq data adjusted for age (continuous), self-reported sex (binary), HIV co-infection (binary; 2 patients not known to be living with HIV but with missing rapid diagnostic tests analyzed as HIV negative), exposure to corticosteroids (binary), and WHO clinical severity classification (categorical) as independent variables. Source data are provided in the Source Data file.

signals. Notably, patients with Delta phase COVID-19 also showed upregulation of pathways related to viral cell entry and trafficking, including receptor-mediated endocytosis, plasma membrane fusion, and syncytium formation.

**Integrated clinicomolecular analyses reveal interferon and TNF-centered immune networks with a coordinated Th1/Th17,**

**endothelial, and profibrotic host response in COVID-19 respiratory failure**

Considering that inter-related biological domains may contribute to the immunopathology of severe COVID-19, we applied hierarchical correlation and network analyses to identify structural relationships between soluble mediators and physiologic indicators of COVID-19 severity and respiratory failure. Among patients with symptomatic

COVID-19 ($N = 240$), hierarchical correlation matrices and force-directed network models identified two groups of strongly correlated mediators centered around IFN-α2/IFN-γ and TNF/lymphotoxin-α, respectively, with IL-15 appearing as a key inter-group mediator (Fig. 5a, b and Supplementary Fig. 5). When HIV status was added as a node, network structure remained consistent (Supplementary Figs. 6,7).

Within each hierarchical correlation matrix and force-directed network, we examined relationships between mediator concentrations, overall morbidity (KPS), and physiologic indicators of acute respiratory failure (oxygen saturation, respiratory rate) (Fig. 5a, b). In addition to reinforcing associations between monocyte/macrophage, NK cell, and Th1-pathway activation and COVID-19 severity, network structures emphasized coordinated Th1/Th17, endothelial, and profibrotic host response in COVID-19 respiratory failure, alongside compensatory, anti-inflammatory immune dampening (Fig. 5a, b). This

included strong relationships between IL-7, IL-15, TGF-α, VEGF-A, MDC and oxygen saturation, with similar associations observed for other closely linked mediators, including lymphotoxin-α, IL-17F, IL-4, and IL-10.

## Unsupervised clustering reinforces prognostic immune signatures in COVID-19 that persist in PLWH and diverge by variant-driven pandemic phase

To further elucidate the presence of prognostic host signatures and their relationship with HIV co-infection and circulating SARS-CoV-2 variants, we applied consensus $k$-means clustering to $\log_{10}$-transformed, scaled, and centered mediator concentrations, excluding clinical variables, to identify immune signatures in patients with symptomatic COVID-19 ($N = 240$). Examination of consensus matrices, cumulative distribution functions, and cluster metrics suggested that a two-cluster (i.e., two COVID-19 Response Signature [CRS]) model was

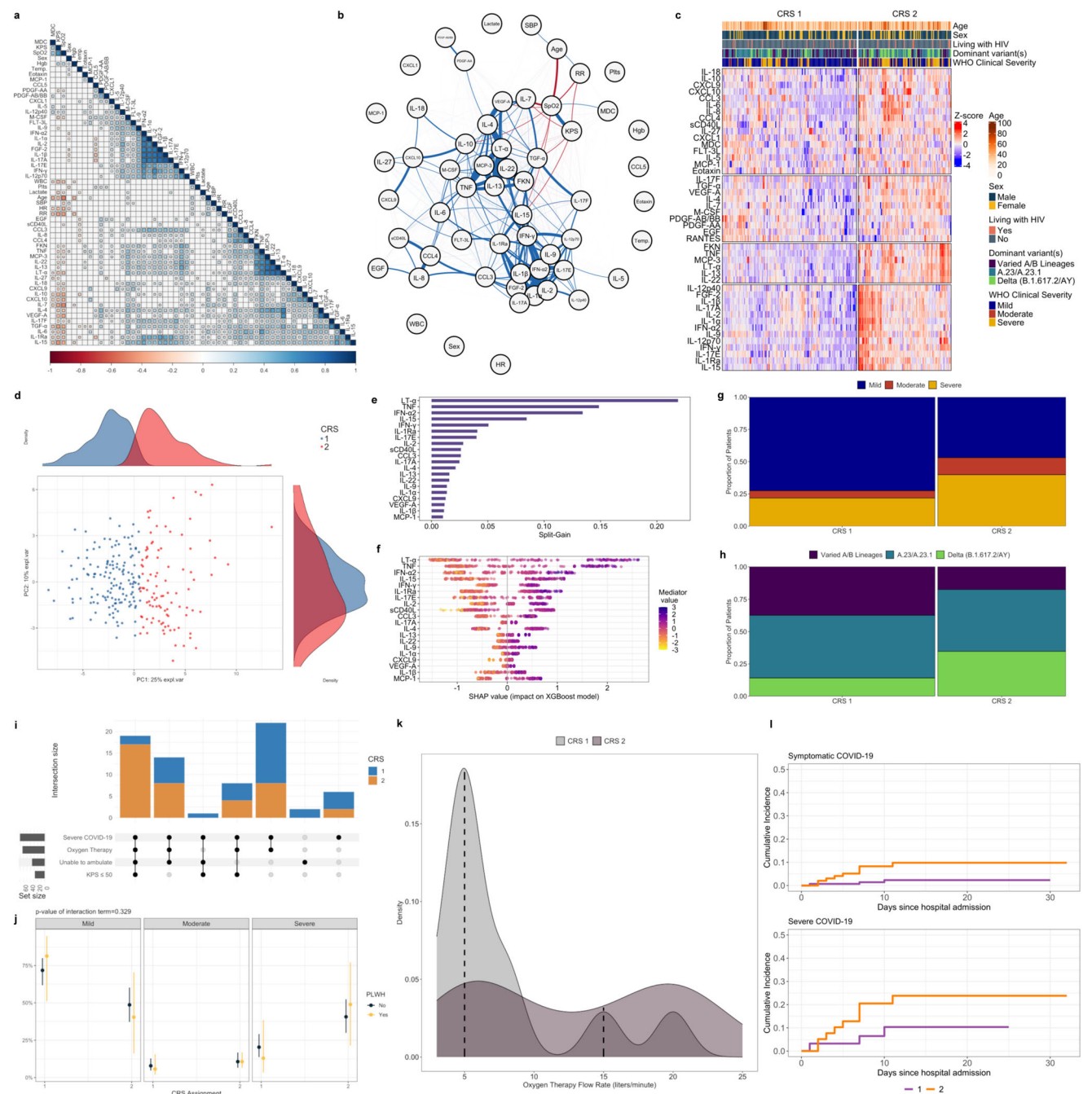

**Fig. 5 | Clinicomolecular profiling of COVID-19 and cluster-derived COVID-19 Response Signatures. a** Hierarchical correlation matrix showing relationships between demographics, clinical variables, and immune mediators in patients with symptomatic COVID-19 ($N = 240$); shaded squares reflect Spearman correlation coefficients filtered by a Benjamini-Hochberg adjusted two-sided $p$ value ≤ 0.05. KPS Karnofsky Performance Status, SpO2 peripheral oxygen saturation, Hgb hemoglobin, Temp temperature, SBP systolic blood pressure, HR heart rate, RR respiratory rate, WBC white blood cell count, Plts platelet count. **b** Force-directed weighted correlation network showing relationships between immune mediators and clinical variables in patients with symptomatic COVID-19 ($N = 240$); network structured on weighted correlations with each variable set as a network node and between-variable correlations significant at a false discovery rate-adjusted two-sided $p$ value ≤ 0.05 indicated by weighted edges (blue edges indicate positive correlation, red edges indicate negative correlation, edge width indicates the strength of correlation based on Spearman coefficient). KPS Karnofsky Performance Status, SpO2 peripheral oxygen saturation, Hgb hemoglobin, Temp temperature, SBP systolic blood pressure, HR heart rate, RR respiratory rate, WBC white blood cell count, Plts platelet count. **c** Heatmap of 45 soluble immune mediator concentrations (values $\log_{10}$-transformed, centered, and scaled) stratified by consensus cluster-derived COVID-19 Response Signatures (CRS; CRS-1, $N = 142$; CRS-2, $N = 98$) in patients with symptomatic COVID-19 ($N = 240$); rows split by $k$-means clustering. Individual patient columns ordered based on differences in mean $z$-score and annotated with age, self-reported sex, HIV co-infection, World Health Organization (WHO) clinical severity classification, and dominant SARS-CoV-2 variant at time of hospitalization. IL-3, GM-CSF, and G-CSF omitted from cluster derivation analysis and heatmap given large proportion of values below the lower limit of quantification. **d** Principal components analysis of immune mediator concentrations in individual patients stratified by cluster-derived COVID-19 Response

Signature (CRS). **e** Importance of immune mediators in gradient-boosted machine classifier for prediction of COVID-19 Response Signature 2 ($N = 98$) vs. 1 ($N = 142$) as per model split-gain values; top 20 variables presented in descending order of importance. **f** Shapley additive explanations (SHAP) values derived from gradient-boosted machine model for prediction of COVID-19 Response Signature 2 ($N = 98$) vs. 1 ($N = 142$); SHAP values > 0 indicate positive impact on prediction while values < 0 indicates negative impact (e.g., high concentrations of lymphotoxin-α have a strong positive contribution to prediction of COVID-19 Response Signature 2). **g, h** Mosaic plots showing distribution of COVID-19 severity and pandemic phases among patients assigned to COVID-19 Response Signature (CRS) 1 ($N = 142$) or 2 ($N = 98$); column width reflects frequencies of patient assignments to each signature. **i** Upset plot showing frequency of severe COVID-19, receipt of oxygen therapy, severely impaired functional status (Karnofsky Performance Status [KPS] ≤ 50), and inability to ambulate among patients assigned to COVID-19 Response Signature (CRS) 1 ($N = 142$) and 2 ($N = 98$). **j** Predicted probabilities of COVID-19 severity classifications stratified by HIV co-infection ($N = 236$; 4 patients without definitive assessment of HIV co-infection excluded); probabilities with 95% confidence interval bars derived from multivariable proportional odds model including WHO clinical severity as ordinal dependent variable and interaction term between COVID-19 Response Signature assignment and HIV co-infection status as independent variable; $p$ value reflects two-sided Wald test of interaction term. **k** Density plot of peak oxygen flow-rates received by patients assigned to COVID-19 Response Signature 1 and 2; dashed lines indicate median oxygen flow-rate for each group ($N = 34$; patients without documented peak oxygen therapy flow-rate excluded). **l** Cumulative incidence of poor hospital outcomes (death or transfer considering discharge alive as a competing risk) for patients with symptomatic ($N = 240$) and severe ($N = 70$) COVID-19 stratified by COVID-19 Response Signature. Source data are provided in the Source Data file.

the optimal fit for the cohort (Fig. 5c, Supplementary Table 10 and Supplementary Fig. 8). Clear between-cluster separation was apparent across the first principal component of mediator variance (Fig. 5d). Compared to CRS-1 ($N = 142$; 59.2%), patients assigned to CRS-2 ($N = 98$; 40.8%) had elevated concentrations of nearly all mediators, reflecting a broad, pro-inflammatory innate and Th1/Th17-dominated host response (Supplementary Table 11). In a gradient-boosted machine classifier model, prediction of CRS-2 assignment was driven by higher concentrations of lymphotoxin-α, TNF, IFN-α, IFN-γ and IL-15, with additional contributions from mediators reflective of monocyte/macrophage (CCL3, CXCL9) and Th1/Th17 pathway (IL-2, IL-17A, IL-17E, IL-22, IL-1α) activation (Fig. 5e, f).

Clinically, patients assigned to CRS-2 exhibited a severe phenotype. This included increased hypoxemic respiratory failure, with more patients requiring oxygen therapy, and among those who did, a significantly higher oxygen requirement (Fig. 5g, i, k, Supplementary Data 4). Cumulative incidence of poor in-hospital outcome (death or higher-level transfer) was significantly higher among patients in CRS-2 (subhazard ratio 4.52 [95% CI 1.23–16.60], $p = 0.023$), a finding that was consistent when models were restricted to patients with severe COVID-19 (subhazard ratio 2.54 [95% CI 0.70–9.16], $p = 0.150$) (Fig. 5l). The proportion of PLWH was similar across CRS 1 and 2 (9.9 and 10.5%, respectively), and there was no evidence that HIV status significantly modified the relationship between CRS and COVID-19 severity risk ($p$ value for interaction = 0.329) (Fig. 5j, Supplementary Data 4).

Significant differences in the distribution of SARS-CoV-2 variant-driven pandemic phases were observed across CRS (Fig. 5h, Supplementary Data 4). Patients assigned to CRS-2 were more frequently hospitalized during the Delta phase, while patients assigned to CRS-1 were mostly hospitalized during that dominated by circulation of imported lineage A and B variants. Patients hospitalized during the A.23/A.23.1 phase were assigned to CRS-1 and 2 in similar proportions. In a multivariable logistic model adjusted for age, self-reported sex, HIV co-infection, pre-enrollment illness duration, and WHO severity classification, the association between Delta phase COVID-19 and CRS-2 assignment persisted (adjusted odds ratio 2.51 [95% CI 1.29–4.89], $p = 0.007$).

## SARS-CoV-2/HIV coinfection is associated with downregulation of pro-inflammatory innate immune pathways, altered protein glycosylation, and reduced type I and II interferon responses

Considering that risk of severe COVID-19 and prognostic host responses appeared generally consistent in PLWH, we directly compared soluble mediator concentrations and gene expression between PLWH and those without HIV, hypothesizing that responses would be conserved. After adjustment for age, self-reported sex, and WHO clinical severity classification, PLWH exhibited significantly lower concentrations of IFN-α2 and multiple mediators of innate myeloid and NK cell activation/chemotaxis, as well as those reflecting Th17 pathway and inflammasome activation. In contrast, PLWH had higher concentrations of IL-5, which, in addition to activating eosinophils, enhances B cell production of mucosal IgA, a key component of the early neutralizing antibody response to SARS-CoV-2 (Fig. 6a, Supplementary Table 12)[27].

GSEA, filtered by an FDR $q$ value ≤ 0.10, reinforced downregulation of key pro-inflammatory innate immune pathways in PLWH, including reduced macrophage and FcR stimulatory pathway activation (Fig. 6b). PLWH also showed reduced O-link protein mannosylation, a process which may affect SARS-CoV-2 entry and trafficking in host cells through changes to viral glycoprotein conformational dynamics[28]. Although associated with FDR $q$ values > 0.10, PLWH also showed reduced responses to IFN-α, IFN-β, and IFN-γ, with broad downregulation of innate immune functions including neutrophil-mediated immunity, inflammasome assembly, TLR4 signaling, and complement activation. In contrast, PWLH showed increased activation of CD8$^+$ T cells, T cell proliferation, and chemotaxis.

## Neutrophil, Th17 pathway, and profibrotic mediators differentiate COVID-19 from the host response to severe influenza and other acute respiratory infections

To determine if immune, endothelial, and growth factor profiles in COVID-19 were distinct or reflected generalized host responses to severe acute respiratory infection (SARI), we analyzed, using our immunoassay panel, cryopreserved serum samples from adults hospitalized with influenza and non-influenza SARI at ERRH from 2017 to 2019 (Supplementary Data 5). We determined the relationship

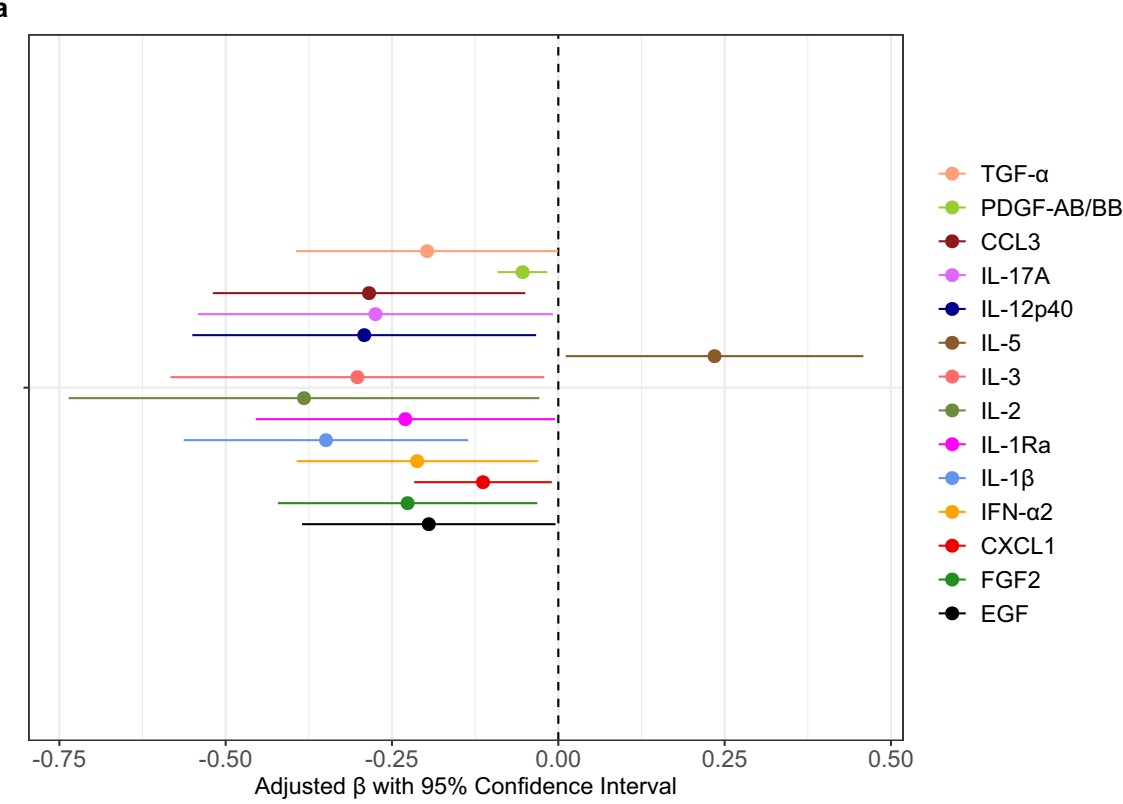

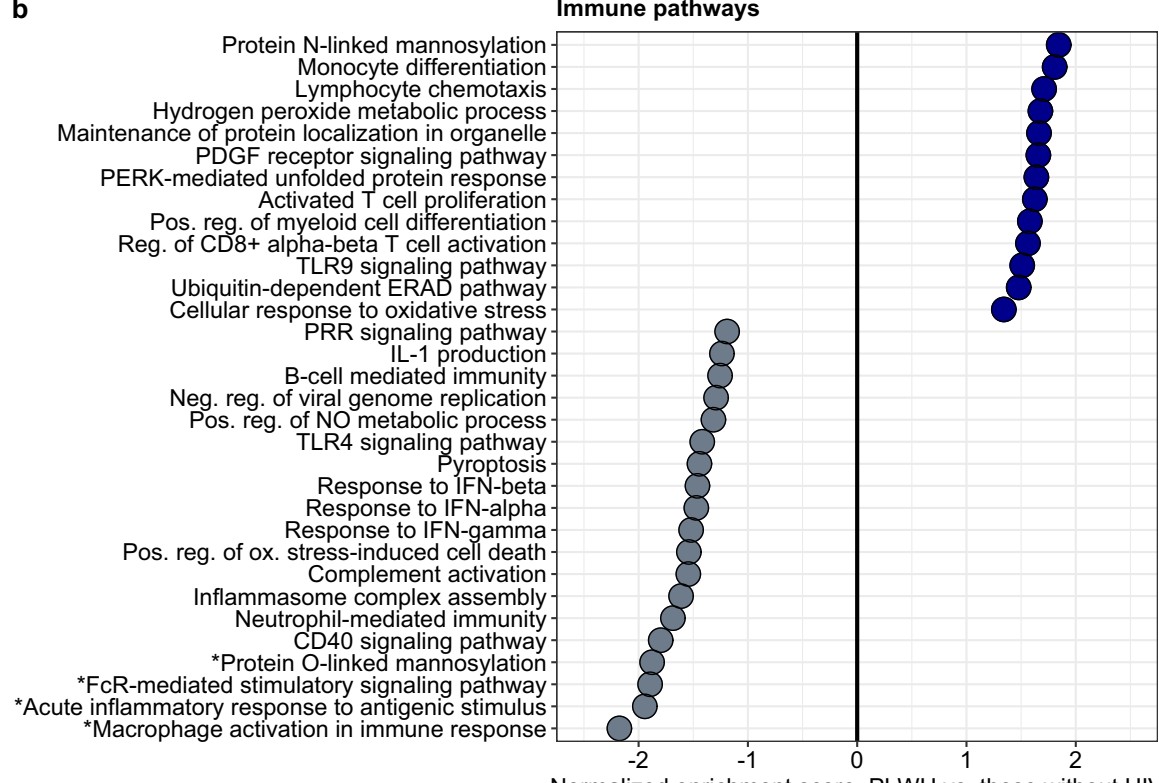

between mediator concentrations and diagnoses of COVID-19 or influenza/non-influenza SARI using multivariable linear regression models, including age, self-reported sex, HIV status, and WHO clinical severity classification. Although concentrations of many mediators were similar across SARI etiology, those reflective of neutrophil chemotaxis and Th17 pathways (IL-8, CCL3, IL-17E) were significantly

higher in patients with COVID-19. COVID-19 patients also showed reduced concentrations of IL-27, which may augment emergency granulopoiesis and neutrophil migration during acute viral respiratory infection (Fig. 7a, b, Supplementary Tables 13, 14)[29]. Consistent with reports of fibrotic lung abnormalities following SARS-CoV-2 infection, patients with COVID-19 had significantly higher concentrations of

**Fig. 6 | Immune mediators and biological pathway enrichment in SARS-CoV-2/HIV co-infection. a** Associations between HIV co-infection and $\log_{10}$-immune mediator concentrations in patients with SARS-CoV-2 infection ($N$ = 302; persons living with HIV, $N$ = 33; patients living without HIV, $N$ = 269; 4 patients without definitive assessment of HIV co-infection excluded); coefficients with 95% confidence interval bars generated in multivariable linear regression models including $\log_{10}$-transformed mediator concentration as dependent variable and HIV co-infection (binary), age (continuous), self-reported sex (binary), and WHO clinical severity classification (categorical) as independent variables. **b** Differential enrichment of key biological pathways among patients living with HIV ($N$ = 16) vs. those without HIV ($N$ = 82; 2 patients without definitive assessment of HIV co-infection excluded); pathway enrichment determined using Gene Set Enrichment Analysis applied to differentially expressed gene sets generated in a DESeq2 model of whole-blood RNAseq data adjusted for age (continuous), self-reported sex (binary), and WHO clinical severity classification (categorical). *Differentially enriched pathways at FDR $q$ value ≤ 0.10; remainder of pathways associated with FDR $q$ value > 0.10. PRR pathogen recognition receptor, NO nitric oxide, ERAD Endoplasmic-reticulum-associated protein degradation. Source data are provided in the Source Data file.

multiple epidermal and platelet-derived growth factors, including TGF-α, EGF, PDGF-AB/BB, and PDGF-AA[30]. As has been reported in HICs, concentrations of IL-6 and other pro-inflammatory mediators were significantly lower in COVID-19 compared to other severe respiratory infections[31]. A gradient-boosted machine classifier model and SHAP metrics reinforced the distinct importance of profibrotic growth factors and neutrophil and Th17 mediators in the SARS-CoV-2 host response, with higher concentrations of EGF, PDGF-AA, FLT-3L, and IL-8, and lower concentrations of IL-27 as top drivers of COVID-19 diagnosis (Fig. 7c, d).

## Discussion

In a nationally-representative prospective cohort study in Uganda, we applied multimodal methods to dissect the immunopathology of COVID-19 across three variant-driven phases of the pandemic. These analyses identified distinct immune signatures associated with COVID-19 severity, some of which reflect underemphasized yet potentially therapeutically exploitable biological pathways. We show that while prognostic host signatures were generally consistent in SARS-CoV-2/HIV co-infection, Delta phase COVID-19 was defined, independent of clinical severity, by exaggerated pro-inflammatory myeloid cell and inflammasome activation and NK and CD8$^+$ T cell depletion. Combining these analyses with a contemporary Ugandan cohort of adults hospitalized with influenza and other severe respiratory infections, we identified activation of epidermal and platelet-derived growth factor pathways as distinguishing features of COVID-19, deepening translational understanding of mechanisms potentially underlying pulmonary fibrosis and other clinical sequelae of SARS-CoV-2 infection.

Over the course of the pandemic, systems biology-based investigations have dissected the host response to SARS-CoV-2 with high resolution, revealing immunopathological signatures associated with COVID-19 severity and potential treatment targets. Consistent with our findings, multiple single-cell and genomic analyses of peripheral blood have identified dysregulated myeloid cell immunity as a key feature of severe COVID-19[8,32,33]. In addition to increased circulation of activated, IFN-stimulated monocytes, emergency granulopoiesis and neutrophil-driven innate immune activation, often accompanied by expansion of aberrant, mature neutrophil subsets, has emerged as a common immunopathological finding in severe COVID-19[8,32,33]. Our multimodal analyses reinforce a distinct role for neutrophil immunity in COVID-19, with neutrophil-driven pathways and neutrophil abundance highly enriched in severe COVID-19, distinguishing this host response to that of influenza and other respiratory infections. Comprehensive single-cell analyses across the COVID-19 severity spectrum have also revealed increased quantities of circulating plasma cells/plasmablasts and IL-15-related exhaustion of circulating NK cells in severe COVID-19, findings consistent with those observed in our cohort[33-35]. Reported T cell signatures in severe COVID-19 have varied, with both extreme T cell activation and T cell lymphopenia, particularly of CD8$^+$ cells, associated with severe illness and poor outcomes[10,36]. Our findings support this profile, with concurrent Th1-pathway upregulation and T cell activation, along with CD8$^+$ T cell depletion, observed in patients with severe COVID-19.

Reinforcing the prognostic relevance of myeloid cell and Th1-driven inflammation among patients with COVID-19 in sub-Saharan Africa, our findings provide locally-relevant biological rationale for use for broad and targeted immunomodulatory therapeutics (e.g., corticosteroids, anti-IL-6 antagonists, JAK/STAT inhibitors) in this setting. Although observational data suggest that corticosteroid administration is associated with improved outcomes for adults with severe COVID-19 in sub-Saharan Africa, use of HIC-derived treatment strategies for sepsis and other severe acute infections have shown no benefit or harm when implemented in the region[14,37,38]. Therefore, establishing pathobiological rationale for use of immunotherapeutics is imperative, especially since some may be associated with context-specific adverse events (e.g., corticosteroid administration without consistent glycemic monitoring or targeted immunosuppression in individuals at high risk for reactivation of latent co-infections)[38].

We identified IL-7, IL-15, and lymphotoxin-α as central mediators underlying severe COVID-19, related respiratory failure, and high-risk immune signatures. Essential for lymphocyte expansion and survival, stromal-derived IL-7 is a strong inducer of memory and effector CD4$^+$/CD8$^+$ T cell proliferation and improves lymphocyte function and metabolic homeostasis[39,40]. IL-7 also influences production of lymphotoxin by lymphoid tissue inducer cells, which in turn induces type I interferons, enhances CD8$^+$ T cell cytotoxicity, and maintains antiviral integrity in secondary lymphoid organs[39,41]. IL-15 has a similarly essential role in development and maintenance of the NK and T cell compartments, particularly for CD8$^+$ T cells[40]. During T cell lymphopenia, a defining feature of severe COVID-19, concentrations of IL-7 increase, likely because of decreased consumption by depleted lymphocytes[39]. However, these physiological increases may be insufficient to adequately replete functional T cell pools, highlighting the restorative potential of recombinant IL-7 administration in severe COVID-19[42]. In contrast, recent studies suggest that prolonged exposure of NK cells to high levels of IL-15 may induce a metabolically exhausted, apoptosis-prone NK cell phenotype in COVID-19[34,43]. Given crucial roles of T and NK cells in modulation of inflammatory responses, generation of immunological memory, and viral clearance, further basic and translational studies are needed to define a role, if any, for IL-7 and IL-15-based therapies that reconstitute and augment T- and NK cell function in COVID-19.

Beyond these specific mediators, our findings deepen in vivo understanding of COVID-19 pathobiology by elucidating a central role for interplay between redox imbalance, innate immune, and microvascular pathways in severe COVID-19. Neutrophil-driven phagocytosis, a key feature of antiviral defense in COVID-19, plays a fundamental role in the generation of reactive oxygen/nitrogen species (ROS/RNS) through respiratory burst, while viral respiratory pathogens are known to inhibit expression of antioxidant transcription factors (e.g., Nrf2)[21,44,45]. Moreover, recent data suggest that production of ROS/RNS and resulting redox imbalance may perpetuate SARS-CoV-2 replication and tissue damage through changes to ER stress and unfolded protein responses, processes we observed to be upregulated in severe COVID-19[46,47]. By inducing alterations to vascular endothelium and glycocalyx function, which were common features of severe COVID-19 in our cohort, ROS/RNS may also be crucial drivers of

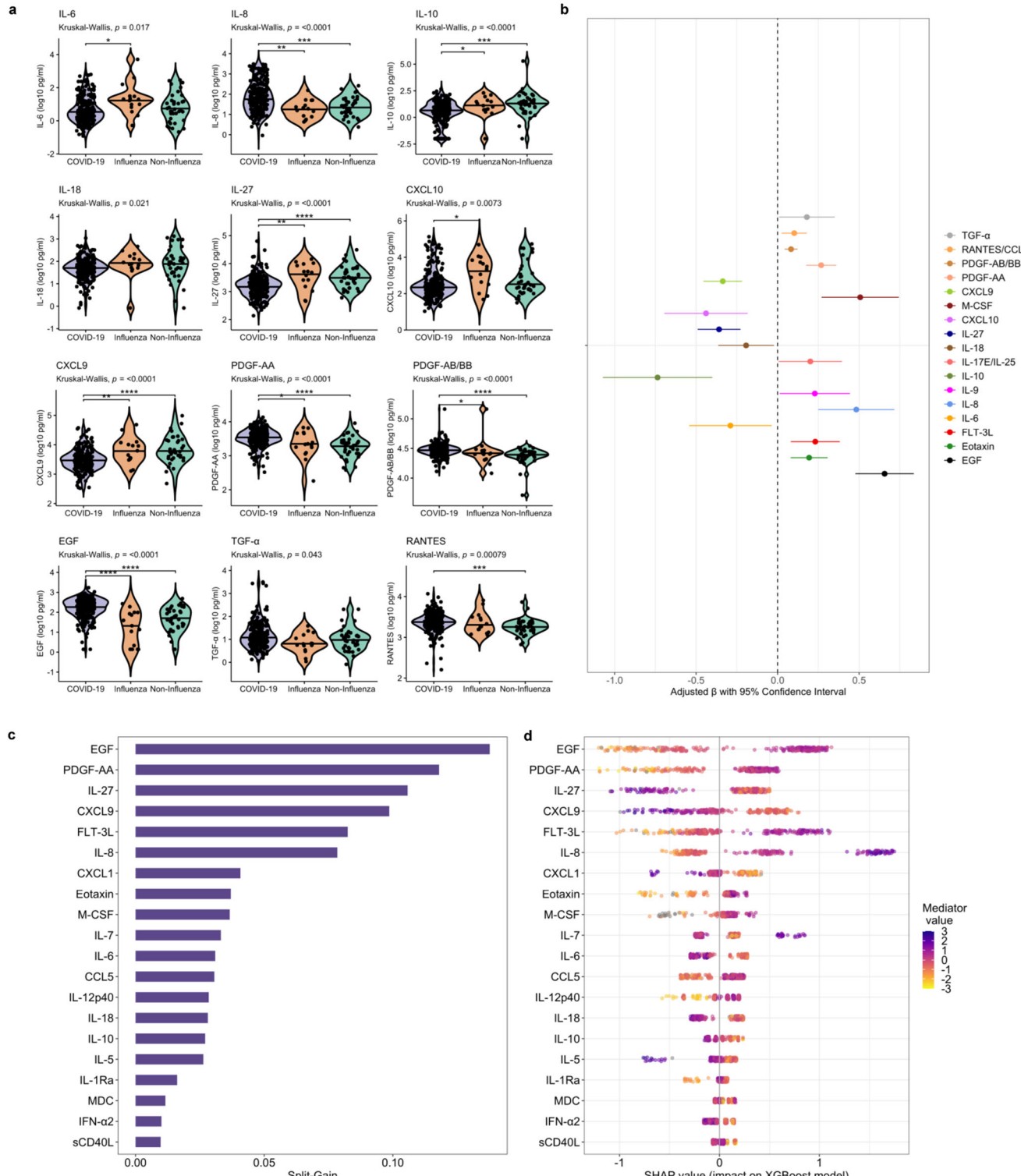

**Fig. 7 | Relationships between immune mediators and severe acute respiratory infection etiology. a** Violin plots showing immune mediator concentrations stratified by severe acute respiratory infection (SARI) etiology ($N = 292$; 66 patients with asymptomatic SARS-CoV-2 infection excluded). Concentrations across groups compared using Kruskal-Wallis H test followed by Dunn's test for multiple comparisons with $p$ values adjusted using Benjamini-Hochberg method; *$p < 0.05$, **$p < 0.01$, ***$p < 0.001$, ****$p < 0.0001$. **b** Associations between COVID-19 and immune mediator concentrations ($N = 292$; 66 patients with asymptomatic SARS-CoV-2 infection excluded); coefficients with 95% confidence interval bars generated in multivariable linear regression models including $\log_{10}$-transformed mediator concentration as dependent variable and SARI etiology (binary; COVID-19 vs. influenza/non-influenza SARI), age (continuous), self-reported sex (binary), HIV co-

infection (binary; 4 patients not known to be living with HIV but with missing rapid diagnostic tests analyzed as HIV negative) and WHO clinical severity classification (categorical) as independent variables. **c** Importance of immune mediators in gradient-boosted machine classifier for prediction of COVID-19 ($N = 240$) vs non-COVID-19 ($N = 52$) SARI as per model split-gain values; top 20 variables presented in descending order of importance. **d** Shapley additive explanations (SHAP) values derived from gradient-boosted machine model for prediction of COVID-19 ($N = 240$) vs non-COVID-19 ($N = 52$) SARI; SHAP values > 0 indicate positive impact on prediction while values < 0 indicates negative impact (e.g., high concentrations of EGF have a strong positive contribution to prediction of COVID-19). Source data are provided in the Source Data file.

microcirculatory dysfunction[48]. Collectively, these observations suggest that ROS/RNS scavengers could complement immunomodulation in severe COVID-19 and support the rationale for clinical trials of these agents.

In vitro experiments in human alveolar epithelial cells suggest that SARS-CoV-2 modulates host protein synthesis, both to enhance viral mRNA translation and inhibit production of antiviral mediators[23]. SARS-CoV-2 nonstructural proteins (e.g., Nsp1) are central to this process, accelerating degradation of host mRNAs and impairing nuclear mRNA export to attain a virally dominated mRNA pool[23,49]. In parallel, Nsp1 preferentially inhibits host translation, including of type I interferons and other antiviral mediators, through blockade of mRNA entry channels and inhibitory binding to ribosomal subunits[50]. Consistent with these findings, we observed evidence of multifaceted impairment of cytoplasmic and mitochondrial protein synthesis in severe COVID-19. While nonstructural proteins of SARS-CoV-2 likely play a key role in infected alveolar epithelial cells, mechanisms underlying this observation in peripheral blood cells are unclear. Although SARS-CoV-2 can infect blood monocytes, we were unable to evaluate this in our samples. As has been reported in other severe viral infections, processes independent of viral cell entry, such as those activated in response to inflammatory or oxidative stress, may have blunted host translation[51]. Nonetheless, considering strong in vitro and in vivo data on the importance of subverted protein synthesis in severe COVID-19, small molecule inhibitors targeting Nsp and related proteins deserve attention as adjunctive anti-viral therapeutics.

Four years into the COVID-19 pandemic, the immunological and clinical impact of HIV co-infection in COVID-19 remain incompletely understood. While key immunological features of severe COVID-19 (e.g., exaggerated innate immune activation, altered type 1 interferon responses, T cell lymphopenia), may be exacerbated by HIV-related immune dysfunction, studies have shown comparable, and in some cases dampened, immune-inflammatory profiles in SARS-CoV-2/HIV co-infection[52–55]. In PLWH in our cohort, most of whom were virologically suppressed and had similar risk of severe COVID-19, prognostic host responses were largely consistent when compared to those without HIV. Interestingly, however, in directly comparative analyses, PLWH showed a relatively diminished innate immune profile, including lower concentrations of IFN-α2 and multiple pro-inflammatory mediators, with upregulation of cytotoxic T cell activation. Despite these differences, higher concentrations of IFN-α2, IL-6, and IL-1, all of which are associated with HIV-related immune activation, differentially scaled with more severe COVID-19 in PLWH. Given the relatively small proportion of PLWH in our cohort, these hypothesis-generating findings should be interpreted with caution. Moving forward, larger-scale immune profiling across the spectrum of HIV-related immunosuppression is needed to establish conserved immunopathological mechanisms in SARS-CoV-2/HIV co-infection, define the role, if any, of synergistic antiviral strategies, and determine if therapeutic targeting of specific immune mediators (e.g., via anti-IL-6 or IL-1 receptor antagonists) has increased benefit in PLWH[16].

Throughout the pandemic, emerging SARS-CoV-2 variants have driven recurrent surges of COVID-19, with global analyses suggesting that COVID-19 severity and mortality were highest during periods dominated by Delta variant circulation[56–58]. In our cohort, patients hospitalized during the Delta variant phase, who were at high risk for severe COVID-19, showed a distinctly dysregulated host profile involving immune, microvascular, and profibrotic growth factor pathways. Moreover, our analyses suggest that observed host responses are not merely a reflection of clinical severity. Even after adjusting for WHO severity classification, among other factors, Delta phase COVID-19 was associated with upregulation of multiple innate immune pathways, particularly inflammasome assembly, as well as NK and CD8+ T cell depletion, ROS generation, and impaired host protein synthesis. These data are consistent with recent in vitro experiments, in which Delta

spike proteins, compared to those of wild-type and G614 variants, triggered greater NF-κB activation in monocytes and pro-inflammatory mediator (IL-1β, IL-6, IL-8, TNF) production in macrophages and dendritic cells[25]. Computational analyses suggest that disproportionate induction of pro-inflammatory mediators in this setting may be related to conformational changes in Delta variant spike RBD, ORF7a, and ORF8 proteins[26]. Patients with Delta phase COVID-19 in our cohort also showed upregulation of plasma membrane fusion, syncytium formation, and endocytosis, which align with in vitro data suggesting that Delta spike proteins exhibit increased fusogenic and endocytic activity[25].

Emerging data suggest that severe SARS-CoV-2 infection, as opposed to influenza, may reprogram macrophages toward profibrotic phenotypes comparable to those found in idiopathic pulmonary fibrosis (IPF)[59]. Consistent with this work, our analyses support a distinct, central, and early role for epidermal and platelet-derived growth factors in the severe COVID-19 host response. This is supported by unique associations between COVID-19 and high concentrations of growth factors implicated in IPF pathogenesis (EGF, PDGF-AB/BB, PDGF-AB/BB, FLT-3L, TGF-α), as well as close relationships between TGF-α, severe COVID-19, and related respiratory failure. As several of these growth factor pathways are targeted by established antifibrotic agents (e.g., nintedanib), our findings reinforce the importance of clinical trials of their early use in COVID-19.

Our study has limitations. First, our study reflects data collected at a single center, albeit one that as a national referral center received patients countrywide with enrollment rates and demographics reflective of national and regional trends. Second, the cross-sectional nature of our sample collection precludes in-depth characterization of temporal host response dynamics, which likely vary during COVID-19[9,34]. Third, although we analyzed immune mediators in a relatively large number of patients, whole-blood RNAseq data were analyzed in a subset. However, demographics, HIV and Delta phase prevalence, and clinical severity in this group were comparable to the larger study population. Fourth, our analyses were limited to peripheral whole blood samples. We were unable to isolate mononuclear cells or perform single-cell RNAseq given resource-limitations at our study site. Our RNAseq subcohort was also not powered for cell type-specific differential gene expression analyses. Fifth, while we divided our study period into pandemic phases using methods similar to those used by the U.S. Centers for Disease Control and Prevention, we were unable to perform patient-level sequencing to confirm variant infections or viral loads[60]. As such, we focused our variant-stratified analyses on patients hospitalized during the Delta phase, during which time up to 90% of nationally sequenced samples were Delta B.1.617.2 and AY sublineages. Our enrollment period also did not include the Omicron variant-dominated phase. Next, most patients enrolled in our study were male. As differences in prognostically-relevant immune responses may be present between males and females with COVID-19, further studies are needed to better define the role of sex and gender in SARS-CoV-2 immunopathology[61]. Also, while we determined corticosteroid exposure for all patients, we were unable to precisely define the timing of administration relative to enrollment, which could have affected immune responses. Lastly, while HIV prevalence in our cohort was comparable to national and regional figures, our HIV-stratified mediator analyses could have been underpowered. As nearly all enrolled PLWH had evidence of virological suppression, immune profiling in more virologically diverse populations of PLWH is needed.

Through multimodal, severity-spanning dissection of SARS-CoV-2 immunopathology in Uganda, we identified distinct immune signatures associated with COVID-19 severity, some of which reflect underrecognized yet therapeutically exploitable pathways. Our findings advance understanding of the COVID-19 host response in the immunologically unique landscape of sub-Saharan Africa, elucidate the impact of circulating variants and SARS-CoV-2/HIV co-infection on

host immunopathology, and strengthen locally relevant rationale for use for broad and targeted immunomodulatory therapeutics in this understudied setting.

## Methods

### Study design and site

The RESERVE-U-C19 study was conducted at ERRH, a 200-bed public hospital in central Uganda. During the study period, ERRH functioned as a national referral center for COVID-19. Patients with SARS-CoV-2 infection nationwide were referred to the facility for management and monitored isolation, the latter for those with asymptomatic infection identified through routine screening at designated surveillance points (e.g., airports, border crossings). Representative of a public COVID-19 treatment unit in sub-Saharan Africa, there was no functional intensive care unit, invasive mechanical ventilation (IMV), or piped oxygen available at ERRH during the study period. Oxygen-therapy was typically provided via nasal cannula and simple or non-rebreathing facemask using concentrators (with devices provided by the RESERVE-U-C19 study program) and/or cylinders. High-flow-nasal-oxygen, continuous-positive-airway-pressure devices, and chest radiography were infrequently available. All management decisions, informed by national COVID-19 treatment guidelines, were made by ERRH clinicians.

### Participants, clinical data collection, and outcomes

Patients were prospectively enrolled in the RESERVE-U-C19 study if they were: (1) admitted to ERRH during the study period, (2) ≥ 5 years of age, (3) had laboratory-confirmed SARS-CoV-2 infection by PCR testing of a naso-/oro-pharyngeal swab sample at a Ministry of Health-accredited laboratory, and (4) were able to provide written informed consent or had a surrogate available to do so. Pregnant females were excluded.

At enrollment, clinical assessments were performed and data were recorded using a modified form developed by the International Severe Acute Respiratory and Emerging Infection Consortium and WHO. For all enrolled patients, rapid testing was performed for *P. falciparum* malaria (SD Bioline Malaria AG P.f. platform; Alere/Abbott, Abbott Park, IL, USA) and HIV (Determine HIV-1/2 Ag/Ab, Alere/Abbott; Chembio HIV 1/2 Stat-Pak, Chembio Diagnostic Systems, Medford, NY, USA, Uni-gold Recombigen HIV-1/2, Trinity Biotech, Ireland). Although patients 5-17 years of age were enrolled in the RESERVE-U-C19 cohort, given variations in normal physiologic parameters, COVID-19 severity criteria, and SARS-CoV-2 host responses across age groups, for this analysis, we included only adults (age ≥ 18 years).

The primary clinical outcome of RESERVE-U-C19 was the frequency of severe COVID-19, defined as the proportion of all enrolled SARS-CoV-2 admissions who fulfilled criteria for severe COVID-19. Based on WHO guidelines, we considered adult patients to have severe COVID-19 if they met ≥ 1 of the following criteria: (1) oxygen saturation ≤ 90%, (2) respiratory rate ≥ 30 breaths/min, (3) showed signs of respiratory distress [chest indrawing, nasal flaring, grunting respirations], and (4) received oxygen-therapy[17]. Case definitions for mild and moderate COVID-19 and asymptomatic SARS-CoV-2 infection are included in the Supplemental Methods. The key secondary study outcome was a composite measure of in-hospital outcome including death in-hospital or transfer to Uganda's highest-level referral hospital (where IMV was available) due to progressive illness severity.

### Sample collection

At enrollment, peripheral blood samples were collected into serum separator tubes and centrifuged, with resulting serum samples stored at −80 °C. In a subset of patients, whole-blood samples were collected in PAXgene blood RNA tubes (cat. no. 762165) (PreAnalytiX, Qiagen/BD, Hombrechtikon, Switzerland) and stored at −80 °C.

### Comparative cohort of patients with influenza and non-influenza SARI

The comparative cohort of patients with influenza and non-influenza SARI included adults (age ≥ 18 years) from the RESERVE-U-1 study, which enrolled patients admitted to ERRH with severe, undifferentiated infection from April 2017 to August 2019. Clinical data and sample collection protocols for RESERVE-U-1 and RESERVE-U-C19 were similar, with residual serum samples for the former stored at −80 °C. As part of the RESERVE-U-1 study, naso-/oro-pharyngeal swab samples were tested via PCR at UVRI for influenza A and B viruses using primers provided by the U.S. Centers for Disease Control and Prevention. Among adults enrolled in RESERVE-U-1, we identified those who fulfilled WHO criteria for SARI (fever [measured temperature ≥ 37.5 °C or reported fever] plus cough of ≤10 days duration with severity sufficient to lead to hospitalization)[62]. We classified patients with influenza and non-influenza SARI into mild, moderate, and severe clinical strata using the criteria described above for COVID-19.

### Serum immunoassays

In cryopreserved serum samples from patients with COVID-19 and influenza and non-influenza SARI, we quantified 48 soluble immune mediators using the Human Cytokine/Chemokine 48-Plex Discovery Assay Array (HD48; Eve Technologies, Calgary, Alberta, Canada; MilliporeSigma, Burlington, MA, USA). Soluble proteins quantified in this Luminex 200-based system (Luminex, Austin, TX, USA) included: sCD40L, EGF, Eotaxin, FGF-2, Flt-3 ligand (FLT-3L), Fractalkine, G-CSF, GM-CSF, GROα/CXCL1, IFN-α2, IFN-γ, IL-1α, IL-1β, IL-1Ra, IL-2, IL-3, IL-4, IL-5, IL-6, IL-7, IL-8, IL-9, IL-10, IL-12p40, IL-12p70, IL-13, IL-15, IL-17A, IL-17E/IL-25, IL-17F, IL-18, IL-22, IL-27, IP-10/CXCL10, MCP-1, MCP-3, M-CSF, MDC (CCL22), MIG/CXCL9, MIP-1α/CCL3, MIP-1β/CCL4, PDGF-AA, PDGF-AB/BB, RANTES, TGF-α, TNF, lymphotoxin-α, and VEGF-A. Samples from different pathogen and severity groups were randomized across plates and analyzed by technicians blinded to pathogen and severity status. Mediator concentrations were quantified in duplicate with the mean used for analysis. Values below the lower limit of assay quantification were replaced with the lowest value that could be reliably quantified for that particular mediator. Values above the upper limit of quantification were replaced with the highest standard curve value for each particular mediator (Supplementary Table 15).

### Whole-blood RNA isolation, library preparation, and sequencing

Total RNA was extracted from whole blood samples in PAXgene blood RNA tubes using PAXgene Blood RNA Kits (cat. no. 762164) (Qiagen, Germantown, MD, USA) in accordance with the manufacturer's protocol (Azenta Life Sciences, South Plainfield, NJ, USA). Total RNA samples were quantified using a Qubit 2.0 Fluorometer (Life Technologies, Carlsbad, CA, USA) and RNA integrity was checked with a 4200 TapeStation (Agilent Technologies, Palo Alto, CA, USA). Samples were initially treated with TURBO DNase (cat. no. AM2238) (Thermo Fisher Scientific, Waltham, MA, USA) to remove DNA contaminants, after which rRNA and globin depletion were performed using QIAseq FastSelect−rRNA HMR and −Globin kits (cat. no. 334375) (Qiagen). RNA sequencing libraries were constructed using the NEBNext Ultra II RNA Library Preparation Kit for Illumina (cat. no. E7770) (New England Biolabs, Inc., Ipswich MA, USA). Sequencing libraries were validated using the Agilent Tapestation 4200 (Agilent Technologies), and quantified using a Qubit 2.0 Fluorometer (ThermoFisher Scientific) as well as by quantitative PCR (KAPA Biosystems, Wilmington, MA, USA). Sequencing libraries were multiplexed, clustered on nine lanes of a flowcell, and loaded on the Illumina HiSeq 4000 instrument according to the manufacturer's instructions (Illumina, Inc., San Diego, CA, USA). Samples were sequenced using a 2 × 150 paired-end configuration. Samples were randomized by severity group prior to RNA extraction, library preparation and sequencing and analyzed by technicians

blinded to severity status. Raw sequence data were converted into FASTQ files and de-multiplexed using Illumina's bcl2fastq 2.17 software. One mis-match was allowed for index sequence identification. Adapters and low-quality bases were trimmed from raw reads using Trimmomatic v0.39[63]. Remaining sequencing reads were aligned to the human genome (GRCh38) using STAR v2.7.10.a, with transcript quantification performed using the quantMode GeneCounts function[64].

## Analysis of soluble immune mediators

To investigate COVID-19 host responses and immune signatures using soluble mediators, we performed a series of supervised regression and classification analyses, in parallel with network correlation, clustering, and PCA. Prior to all analyses, mediator concentrations were $log_{10}$ transformed to minimize skew, with resulting distributions assessed visually using histograms and Q-Q plots.

Concentrations of soluble mediators stratified by COVID-19 severity were visualized using standardized heatmaps (*ComplexHeatmap* R package). To determine the relationship between soluble mediator concentrations and COVID-19 severity (asymptomatic SARS-CoV-2 infection, mild, moderate, severe COVID-19) we applied Kruskal-Wallis tests followed by Dunn's test for multiple comparisons (*ggpubr, rstatix* R packages) and generated BH-adjusted multivariable regression models including COVID-19 severity as an ordinal dependent variable (*ordinal, sjPlot, jtools* R packages). To determine mediators most important in predicting severe COVID-19, we performed feature selection using gradient-boosted machine classifier models (*xgboost, caret* R packages). Select model hyperparameters (learning rate [eta], tree depth [max_depth], and number of trees [nround]) were tuned using tenfold cross validation, with remaining hyperparameters left at default settings. We identified the most important predictive mediators using their respective split-gain values and their contribution to the classifier model using Shapley Additive Explanation (SHAP) values. We evaluated the relationship between mediator concentrations and reported duration of illness at enrollment using robust regression, with patient-level datapoints stratified by COVID-19 severity. We compared mediator concentrations by COVID-19 severity in symptomatic patients enrolled within 7 days of illness using Wilcoxon rank sum tests. Where specified, we used similar methods to those described above (Kruskal-Wallis and Dunn's tests, multivariable regression models, hyperparameter tuned gradient-boosted machine classifiers and SHAP values) to determine the relationship between soluble mediator concentrations, HIV co-infection, and hospitalization during different variant-driven pandemic phases.

To elucidate immune signatures in symptomatic COVID-19 patients using unsupervised methods, we performed hierarchical and force-directed network correlation, clustering, and PCA, inclusive and agnostic of clinical data. First, to explore the structure of relationships between mediators and indicators of COVID-19 severity and respiratory failure, we generated BH-adjusted hierarchical and force-directed weighted correlation networks (*corrplot, qgraph, fdrtool* R packages). For the latter, which were based on the Fruchterman-Reingold method, mediator and clinical variables were set as network nodes with between-node correlations significant at a FDR-adjusted $p$ value ≤ 0.05 indicated by weighted edges[65,66]. To identify central nodes around which networks may be coordinated, we calculated node strength and expected influence, with higher values of these metrics indicating that a node is involved in large number of interactions, in this case, correlation weights.

To identify COVID-19 immune signatures agnostic of clinical data, we applied consensus $k$-means clustering to $log_{10}$-transformed, scaled, and centered mediator concentrations (*ConsensusClusterPlus* R package). Using Euclidean distance and a range of clusters ($k$) set from $k = 2$ to $k = 12$, the consensus cluster algorithm was run over 1000 subsamples, with item and feature sampling set at 0.8 and 1, respectively. We determined the optimal cluster (i.e., signature) partition through

inspection of consensus matrices and cumulative distribution plots, which we confirmed using over 20 indices of cluster stability and validity (*NbClust* R package). We visualized patient-level, between-cluster variance in mediator concentrations using PCA (*FactoMineR, factoextra, PLSDAbatch* R packages) and standardized heatmaps (*ComplexHeatmap* R package). Clinical severity indicators across signatures were visualized using upset plots (*ComplexUpset R package*). Differences in hospital outcomes were analyzed using cumulative incidence functions and competing risks regression (*tidycmprsk, ggsurvfit*, R packages).

## Analysis of whole-blood RNAseq data and digital cytometry deconvolution

Independent of immune mediator analyses, we analyzed whole-blood RNAseq data from a subset of enrolled COVID-19 patients. After removing reads with low abundance (<10 counts), we performed covariable adjusted differential gene expression analysis to identify genes that were differentially expressed between key groups of interest based on $log_2$-fold change and BH-adjusted $p$ values (*DESeq2* R package). Between-group comparisons included: patients with severe and non-severe COVID-19 (adjusted for age, self-reported sex, and HIV co-infection), patients admitted during and prior to the Delta variant phase (adjusted for age, self-reported sex, HIV co-infection, corticosteroid exposure, and WHO clinical severity classification), and patients with and without HIV co-infection (adjusted for age, self-reported sex, and WHO clinical severity classification). Following each comparison, differentially expressed gene sets (ranked by BH-adjusted $p$ values) were subjected to GSEA using the Biological Process Gene Ontology resource (MSigDB; GO biological process, 7763 gene sets) to infer biological pathway enrichment across respective groups[67,68]. Enriched gene sets were manually reviewed, and those presented were included to highlight enrichment of pathologically-important and diverse pathways. To infer overall abundance of immune cell subsets using bulk RNAseq data, we performed digital cytometry deconvolution via the CIBERTSORTx platform (absolute mode) and LM22 reference matrix[69]. Absolute immune cell abundances across groups were analyzed using PCA, Wilcoxon rank sum tests, and multivariable regression models as indicated.

## Comparative analyses of patients with COVID-19 and influenza and non-influenza SARI

To determine the relationship between soluble mediator concentrations and SARI etiology (symptomatic COVID-19, influenza, non-influenza SARI) we applied Kruskal-Wallis tests followed by Dunn's test for multiple comparisons (*ggpubr, rstatix* R packages), as well as multivariable linear regression models adjusted for age, self-reported sex, HIV co-infection, and WHO clinical severity classification (*jtools, sjplot* R packages). To determine mediators most important in predicting COVID-19 (vs. influenza or non-influenza SARI), we performed feature selection using hyperparameter-tuned gradient-boosted machine classifier models as described above.

## Missing clinical and laboratory data

For symptomatic COVID-19 patients, missing clinical and laboratory data were multiply imputed using chained equations and predictive mean matching (*mice* R package) (Supplementary Table 16). Five imputed datasets were reviewed for convergence and plausibility after which one was randomly selected for use.

## HIV-1 viral load testing

For study purposes, HIV-1 viral loads were quantitated in residual cryopreserved serum samples at the Columbia University Center for Advanced Laboratory Medicine. Viral loads were quantitated using the cobas HIV-1 assay via the cobas 6800 System (Roche Diagnostics, Basel, Switzerland), which has a linear range of detection of 20 to

$1.0 \times 10^7$ copies/ml. Due to limited residual sample volumes, samples were diluted 2–5-fold with molecular grade water to obtain the required testing volume for the assay. As such, we designated patients as having suppressed viral loads based on limits of detection ranging from 40 to 100 copies/ml.

## Reporting summary

Further information on research design is available in the Nature Portfolio Reporting Summary linked to this article.

## Data availability

Raw bead-based proteomics data have been deposited in Dryad under accession code https://doi.org/10.5061/dryad.2bvq83bvd (https://datadryad.org/stash/dataset/doi:10.5061/dryad.2bvq83bvd). Raw RNAseq data have been deposited in NIH/NCBI Sequence Read Archive and dbGaP under accession numbers PRJNA950542 (https://www.ncbi.nlm.nih.gov/bioproject?LinkName=sra_bioproject&from_uid=27274595) and phs003246.v1.p1 (https://www.ncbi.nlm.nih.gov/projects/gap/cgi-bin/study.cgi?study_id=phs003246.v1.p1), respectively. In concordance with participant consent and institutional certification of genomic data sharing, raw RNAseq data are available to investigators with an IRB-approved protocol. Requests for access can be made to the NIH/NIAID Data Access Committee (niaid_data-sharing@niaid.nih.gov). All other data are available in the article and its Supplementary files or from the corresponding author upon request. Source data are provided with this paper.

## Code availability

Analytic code is available in GitHub at https://github.com/mjc2244/Multimodal-immune-profiling-of-SARS-CoV-2-in-Uganda and https://github.com/mastorkia/bulkRNAseq_COVID-Uganda.

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

## Acknowledgements

This work was supported by the National Institute of Allergy and Infectious Diseases (K23AI163364 and R21AI171249 to M.J.C.), the Fogarty International Center MakCHS-Berkeley-Yale Pulmonary Complications of AIDS Research Training (PART) Program (D43TW009607, sub-award to B.B.), and the National Center for Advancing Translational Sciences (UL1TR001873, sub-award to M.R.O.), National Institutes of Health. Additional support was provided by Burroughs Wellcome Fund/American Society of Tropical Medicine and Hygiene (M.J.C.). The authors would like to thank the patients enrolled in this study, their families, and their fellow clinicians and scientists for providing outstanding clinical care and laboratory work despite considerable personal risk.

## Author contributions

MJC, BB, and MRO conceived the study and its design. MJC, BB, JJL, NO, J. Kayiwa, J. Kiconco, MM, CN, ER, IN, S. Kyebambe, MH, BW, and BK collected, organized, and entered clinical data and performed laboratory work. MJC, XC, and MA performed bioinformatic and statistical analyses. MJC, BB, XC, MA, TP, HKB, FO, S. Kisaka, NK, TP, AT, WIL, and MRO contributed to data analysis and interpretation. MJC drafted the manuscript. All authors critically revised the drafted manuscript and approved of the final manuscript.

## Competing interests

M.J.C. and M.R.O'D. were investigators for clinical trials evaluating the efficacy and safety of remdesivir, convalescent plasma, and anti-SARS-

CoV-2 hyperimmune globulin in hospitalized patients with COVID-19, sponsored by Gilead Sciences, Amazon, and the National Institutes of Health, respectively. Compensation for this work was paid to Columbia University. The remaining authors declare no competing interests.

### Ethical approval
Each enrolled participant or their surrogate provided written informed consent. Study protocols were approved by ethics committees at Columbia University (AAAR1450), Uganda Virus Research Institute (GC/127/17/02-06/582), and Uganda National Council for Science and Technology (HS2308).

### Additional information

**Matthew J. Cummings** [1,2] ✉, **Barnabas Bakamutumaho**[3], **Julius J. Lutwama**[3], **Nicholas Owor**[3], **Xiaoyu Che**[2,4], **Maider Astorkia**[2], **Thomas S. Postler** [5], **John Kayiwa**[3], **Jocelyn Kiconco**[3], **Moses Muwanga**[6], **Christopher Nsereko**[6], **Emmanuel Rwamutwe**[6], **Irene Nayiga**[6], **Stephen Kyebambe**[6], **Mercy Haumba**[3], **Henry Kyobe Bosa** [7,8], **Felix Ocom**[8], **Benjamin Watyaba** [9], **Bernard Kikaire** [9,10], **Alin S. Tomoiaga**[1,11], **Stevens Kisaka**[12,13], **Noah Kiwanuka**[12], **W. Ian Lipkin** [2,14,15], **Max R. O'Donnell**[1,2,15], **Collaboration for Clinical and Laboratory Characterization of COVID-19 in Uganda\***

[1]Division of Pulmonary, Allergy, and Critical Care Medicine, Department of Medicine, Vagelos College of Physicians and Surgeons, Columbia University, New York, NY, USA. [2]Center for Infection and Immunity, Mailman School of Public Health, Columbia University, New York, NY, USA. [3]Department of Arbovirology, Emerging and Re-emerging Infectious Diseases, Uganda Virus Research Institute, Entebbe, Uganda. [4]Department of Biostatistics, Mailman School of Public Health, Columbia University, New York, NY, USA. [5]Department of Microbiology and Immunology, Vagelos College of Physicians and Surgeons, Columbia University, New York, NY, USA. [6]Entebbe Regional Referral Hospital, Entebbe, Uganda. [7]Uganda Peoples' Defence Forces, Kampala, Uganda. [8]Ministry of Health, Kampala, Uganda. [9]European and Developing Countries Clinical Trials Partnership-Eastern Africa Consortium for Clinical Research, Uganda Virus Research Institute, Entebbe, Uganda. [10]Department of Pediatrics, Makerere University College of Health Sciences, Kampala, Uganda. [11]Department of Accounting, Business Analytics, Computer Information Systems, and Law, Manhattan College, New York, NY, USA. [12]Department of Epidemiology and Biostatistics, Makerere University School of Public Health, Kampala, Uganda. [13]Institute of Tropical and Infectious Diseases, University of Nairobi, Nairobi, Kenya. [14]Department of Pathology and Cell Biology, Vagelos College of Physicians and Surgeons, Columbia University, New York, NY, USA. [15]Department of Epidemiology, Mailman School of Public Health, Columbia University, New York, NY, USA. \*A list of authors and their affiliations appears at the end of the paper. ✉e-mail: mjc2244@columbia.edu

## Collaboration for Clinical and Laboratory Characterization of COVID-19 in Uganda

**Matthew J. Cummings** [1,2] ✉, **Barnabas Bakamutumaho**[3], **Julius J. Lutwama**[3], **Nicholas Owor**[3], **John Kayiwa**[3], **Jocelyn Kiconco**[3], **Moses Muwanga**[6], **Christopher Nsereko**[6], **Emmanuel Rwamutwe**[6], **Irene Nayiga**[6], **Stephen Kyebambe**[6], **Mercy Haumba**[3], **Henry Kyobe Bosa** [7,8], **Felix Ocom**[8], **Benjamin Watyaba**[3], **Bernard Kikaire**[3] & **Max R. O'Donnell**[1,2]

A full list of members and their affiliations appears in the Supplementary Information.

