## [Peer Review File · Nature Communications]

REVIEWER COMMENTS

Reviewer #1 (Remarks to the Author):

The authors present in their well written manuscript a comprehensive study of the serum proteome and transcriptome in patients with Covid-19 in Uganda. A subgroup of these patients were coinfecting with HIV-1. They correlate their results to the severity of disease, infection with different SARS-CoV-2 variants and they compare the findings to Covid-19 patients with concomitant HIV-1 infections and to patients infected with influenza or other respiratory infections. The findings have important implications for the understanding of the pathophysiology of Covid-19 and for the rationale of immunomodulatory interventions.

The manuscript is well written. Although the limit of references has been reached it would have been appropriate to discuss the results in comparison to the results of other omics analyses by other groups.

There were only a few points to address:

Line 253: The abbreviation (PCa should be explained at first appearance (not at second appearance in line 711)

Heat maps 1b and 5c: What is the order of the patients (horizontal values)? Overall density of the Z scores?

Error in Figure 4d: indicated legend d presumably is c as c is lacking.

Error in Figure 7: the indicated figure 7e heatmap is missing.

Table S5: It is unclear to which abundance is referring: to asymptomatic Covid 19 patients or to healthy controls (which are not indicated). This should be explained.

Reviewer #2 (Remarks to the Author)

Using circulating protein profiling (of 48 select markers) and blood transcriptomics (of a subset of patients N=100), Cummings and colleagues assess the immune profiles of a COVID-19 adult cohort from Uganda (N=306). Patients span the disease severity spectrum from asymptomatic to severe disease with fatal outcomes. Three aspects of this study are of potential interest. First, the analysis of HIV+ vs. HIV- patients, although the numbers are on the low side and HIV status itself is not a strong correlate of disease severity, thus the signal that emerges are weak at best. Second, this study covers a patient cohort from sub-Saharan Africa, which is relatively understudied using immune profiling approaches. However, without comparative analysis with other populations, it is not clear whether any population specific immune response characteristics were uncovered here. For example, the disease severity

correlates emerged are largely consistent with those reported previously in other populations. Third, the authors also conducted a comparative analysis of COVID-19 with other contemporaneous respiratory infection patients in the same region. The study design is a strength, however, the presence of many potential confounding variables including medications and timing (see also below) and the relatively small sample sizes limit the immunological insights that can be obtained. For example, ref. 2 should be an informative dataset to help assess some of the differences further in conjunction with the rich COVID-19 data sets available.

Thus, while I applaud the authors' efforts, a major limitation of this manuscript is the lack of conceptual and biological advances and insights, particularly given the extensive immune profiling data and observations already available on COVID-19 with 3+ years of intensive studies by many groups. The authors also did not cite many of the pioneering immune profiling papers, particularly those using and integrating circulating protein, multimodal single cell, transcriptomic, and metabolomic analyses that reported similar findings (e.g., see select references below – the authors should do a more thorough survey to ensure the key papers are cited). Another concern is the cross-sectional nature of the analysis without longitudinal information. The systemic immune response to COVID-19 is highly dynamic across many variables. At the minimum, time since symptom onset should be included as a co-variate (e.g., the authors have also shown in fig 3a that some of the circulating proteins they examine are associated with time across patients.) Without longitudinal data it is not clear whether any of the group-based comparisons are confounded by response kinetics. For example, type I IFNs and downstream responses diminish quickly after peaking in blood early (line 178, see refs. 1 and 4 below). Similarly, while IL-15 levels can persist in the severe patients, IL-15 and associated transcriptional responses such as those in NK cells can be highly dynamic and dependent on disease severity (ref. 4 below). The same concerns apply to the comparison among COVID-19, influenza, and other respiratory infections (see ref. 2 below).

Other points:

“Escalating severity” – this term was used in a few places in the manuscript. It implies temporal changes, but I think the authors meant to suggest associative changes between severe and less severe groups. Please clarify and be more specific.

A recurrent question I have throughout the manuscript is the extent by which the reported circulating protein marker or transcriptomic correlates are consistent with those reported by others. It is important to put findings into context, particularly in comparing to other populations/regions/countries including those from other regions of Africa.

How was the subset of 100 patients chosen for transcriptomic profiling?

In some of the analyses, a comparison was made between severe patients and the rest. Why not use the entire disease severity spectrum?

Line 215: “cell lines” – those are not cell lines.

Line 218: not clear what “reducing thymic selection” meant here, particularly given that blood, not thymic tissues, was used. In general, the manuscript tends to list many GSEA based gene set labels that may or may not reflect the underlying complex biology being captured by cell states or circulating protein levels in blood. The authors should go over and interpret these carefully, e.g., by using single cell data available in the public domain to delineate where the bulk signals might be coming from.

Line 240 and on: the broad downregulation of translational, ribosomal, and other biosynthetic processes has been reported in responses to viral infections including COVID-19. Anything specific to this population? Is this linked to the IFN related down-regulation of biosynthetic capacity to counter viral replication?

Overall, the cohort is quite male bias. It would be good to include a discussion on the caveats and implications of this bias.

Reference:

1. Arunachalam, P.S., Wimmers, F., Mok, C.K.P., Perera, R.A.P.M., Scott, M., Hagan, T., Sigal, N., Feng, Y., Bristow, L., Tsang, O.T.-Y., et al. (2020). Systems biological assessment of immunity to mild versus severe COVID-19 infection in humans. *Science*.
2. Dunning, J., Blankley, S., Hoang, L.T., Cox, M., Graham, C.M., James, P.L., Bloom, C.I., Chaussabel, D., Banchereau, J., Brett, S.J., et al. (2018). Progression of whole-blood transcriptional signatures from interferon-induced to neutrophil-associated patterns in severe influenza. *Nat. Immunol.* 19, 625–635.
3. Kuri-Cervantes, L., Pampena, M.B., Meng, W., Rosenfeld, A.M., Ittner, C.A.G., Weisman, A.R., Agyekum, R.S., Mathew, D., Baxter, A.E., Vella, L.A., et al. (2020). Comprehensive mapping of immune perturbations associated with severe COVID-19. *Sci. Immunol.* 5.
4. Liu, C., Martins, A.J., Lau, W.W., Rachmaninoff, N., Chen, J., Imberti, L., Mostaghimi, D., Fink, D.L., Burbelo, P.D., Dobbs, K., et al. (2021). Time-resolved systems immunology reveals a late juncture linked to fatal COVID-19. *Cell* 184, 1836–1857.e22.
5. Mathew, D., Giles, J.R., Baxter, A.E., Oldridge, D.A., Greenplate, A.R., Wu, J.E., Alanio, C., Kuri-Cervantes, L., Pampena, M.B., D'Andrea, K., et al. (2020). Deep immune profiling of COVID-19 patients reveals distinct immunotypes with therapeutic implications. *Science*.
6. Schulte-Schrepping, J., Reusch, N., Paclik, D., Baßler, K., Schlickeiser, S., Zhang, B., Krämer, B., Krammer, T., Brumhard, S., Bonaguro, L., et al. (2020). Severe COVID-19 Is Marked by a Dysregulated Myeloid Cell Compartment. *Cell*.

Reviewer #3 (Remarks to the Author):

In this paper, the team provides a comprehensive plasma and whole blood transcriptomic analysis of a cohort of individuals with SARS-CoV-2 infection in Uganda. The report is well written and significantly adds to the literature with the clear characterization of the immunophenotype in this population which includes ~11% prevalence of concurrent HIV infection. The authors further add a comparison to individuals with influenza and other severe acute respiratory infections which adds specificity to their findings about COVID-19 pathogenesis. Overall, the paper adds key data on the pathogenesis of this infection outside of the high income country setting which is largely consistent with prior findings, but raises some questions for further exploration.

1. A few additional details about the cohort would help interpretation of the findings:

-Methods description of how asymptomatic patients were identified – routine screening, testing at hospital presentation for other reasons etc

-Timing from onset of symptoms to sample collection – these are explored in Table S2 and Figure 3, but should be included in the demographics table.

-Timing from use of corticosteroids to sampling (pre or post sampling)

-Separate table defining the characteristics of the subset of individuals included in the RNAseq experiments, currently text description suggests similar structure of distribution, but this should be presented in a table.

2. From a top level, it is not clear that the paper significantly addresses metabolism. The findings related to metabolic pathways are primarily around phagocytic respiratory burst and reactive oxygen species generation, which are likely related to neutrophil granule composition/degranulation. These are not per se related to the global regulation of immune cell metabolism as typically referenced to describe respiratory capacity in immune cell subsets. Would consider modifying that reference in the title.

3. The analysis of the subgroup of participants with HIV is somewhat limited by the very low number of PLWH in the severe group (n=4) and some of these data seem to be overinterpreted. As the authors correctly identify, the small group and large proportion with fully suppressed viremia makes it difficult to conclude that this sample definitively answers questions about the role of HIV in SARS-CoV-2 responses and would suggest that the authors consider a few points to modify their data presentation or interpretation:

Figure 1 - the models of proportional probabilities seem are limited by the very low numbers of individuals in the HIV positive group (severe with only 4) as suggested by the wide confidence intervals. The data presented in supplemental figure s2 are more clear, and would suggest either moving the subgroup severity analyses to supplemental or amending the text to indicate that these analyses have very limited power.

Figure 3b – It appears that the reported p values here are the nominal p values and not the BH-adjusted ones in Supplemental Table s6. This should be clearly indicated as the other plots have presented adjusted p values. While this does mean that some of the reported comparisons are not significant in the fully adjusted model, I agree with the authors that the results are consistent with the overall cohort and the time course models in Figure 3a support this conclusion. But the lack of adjustment should be indicated.

In the paragraph discussing the comparison of PLWH to the overall cohort, the text should be clarified to indicate that the majority of pathways in Figure 6b have FDR q values >0.1 and any associations are exploratory (currently written as “Beyond FDR q value” which is ambiguous).

Overall, the data on PLWH is very valuable, but with very limited numbers and the presentation and discussion should reflect this.

4. The section on disruption of protein synthesis suggests that impaired transcription and cytosolic mitochondrial translation in patients with severe COVID-19 may be related directly to SARS-CoV-2 subversion of protein synthesis. The data cited to support this are from human lung alveolar cell lines that are directly infected. It is unclear how this would relate to a global immune response profile in which the vast minority of cells are likely directly exposed to or infected by SARS-CoV-2. To suggest that the peripheral immune profile is in support of viral entry, the authors would need to demonstrate that this signature is linked to some measure of virus activity in the plasma or in these cells. page 10 lines 237-247. Would consider modifying this in the results and discussion or considering an alternate explanation of how infection and global inflammation may lead to a disruption in these pathways that is not directly dependent on infection of the cells with SARS-CoV-2.

Minor comments

References 15-18 which provide the data supporting high mortality rates in sub Saharan Africa are not cited in the text where they should be (line 111 which currently only has ref 13,14 listed). It is also of note that differences in mortality may reflect not a unique pattern of inflammation but rather resource constraints and this point is not clearly discussed.

The COVID-19 cohort is 75% male, which should be noted as a limitation, given the association of male sex and/or gender with risk of mortality. Although the models are adjusted, minimal representation of females may limit the ability to identify relevant differences.

In the section on the NLRP3 inflammasome, a discussion of the specific leading edge genes in the inflammasome pathway that lead to this pathway enrichment would be of interest (Lines 320-333).

In Figure 5, panels g,i,k,l are somewhat redundant (i.e. oxygen use is part of the definition of severity) as they are all ways of demonstrating the enrichment of severe phenotypes in the CRS-2 group. Would consider choosing one way of presenting these data.

Reviewer #1

The authors present in their well written manuscript a comprehensive study of the serum proteome and transcriptome in patients with Covid-19 in Uganda. A subgroup of these patients were coinfecting with HIV-1. They correlate their results to the severity of disease, infection with different SARS-CoV-2 variants and they compare the findings to Covid-19 patients with concomitant HIV-1 infections and to patients infected with influenza and other respiratory infections. The findings have important implications for the understanding of the pathophysiology of Covid-19 and for the rationale of immunomodulatory infections.

We thank the reviewer for this comment.

The manuscript is well written. Although the limit of references has been reached it would have been appropriate to discuss the results in comparison to the results of other omics analyses by other groups.

We thank the reviewer for this comment. We have added a new paragraph to the discussion section comparing our main results with those of single-cell and genomic analyses of the COVID-19 host response with the relevant references added (Lines 458-476; see below). We have also emphasized these comparisons throughout multiple other sections of the discussion.

“Over the course of the pandemic, systems biology-based investigations have dissected the host response to SARS-CoV-2 with high resolution, revealing immunopathological signatures associated with COVID-19 severity and potential treatment targets. Consistent with our findings, multiple single-cell and genomic analyses of peripheral blood have identified dysregulated myeloid cell immunity as a key feature of severe COVID-19 [8,32,33]. In addition to increased circulation of activated, IFN-stimulated monocytes, emergency granulopoiesis and neutrophil-driven innate immune activation, often accompanied by expansion of aberrant, mature neutrophil subsets, has emerged as a common immunopathological finding in severe COVID-19 [8,32,33]. Our multimodal analyses reinforce a distinct role for neutrophil immunity in COVID-19 pathobiology, with neutrophil-driven pathways and neutrophil abundance highly enriched in severe COVID-19 and distinguishing this host response to that of influenza and other respiratory infections. Comprehensive single-cell analyses across the COVID-19 severity spectrum have also revealed increased quantities of circulating plasma cells/plasmablasts and IL-15-related exhaustion of circulating NK-cells in severe COVID-19, findings consistent with those observed in our cohort [33-35]. Reported T-cell signatures in severe COVID-19 have varied, with both extreme T-cell activation and T-cell lymphopenia, particularly of CD8+ cells, associated with severe illness and poor outcomes [10,36]. Our findings support this profile, with concurrent Th1-pathway upregulation and T-cell activation, along with CD8+ T-cell depletion, observed in patients with severe COVID-19.”

Line 253: The abbreviation PCA should be explained at first appearance (not at second appearance in line 711).

We have made the suggested edit (Line 251).

Heat maps 1b and 5c: What is the order of the patients (horizontal values)? Overall density of the Z scores?

We thank the reviewer for the opportunity to clarify this point. Individual patients were first stratified by WHO clinical severity (Figure 1b) and COVID-19 Response Signature (Figure 5c). Within each of these strata, as per the ComplexHeatmap R package, individual patient columns were ordered based on differences in mean z-scores. We have added text to the legends for Figures 1b (Lines 1090-1091) and 5c (Lines 1177-1178) to clarify this.

Error in Figure 4d: indicated legend d presumably is c as c is lacking.

We thank the reviewer for this observation. We have corrected the Figure 4 legend (Line 1145).

Error in Figure 7: the indicated figure 7e heatmap is missing.

We thank the reviewer for this observation. The Figure 7e heatmap was removed at a prior stage of manuscript preparation but the legend remained. We have now removed the Figure 7e legend.

Table S5: It is unclear to which abundance is referring: to asymptomatic Covid 19 patients or to healthy controls (which are not indicated). This should be explained.

Table S6 presents the abundance of immune cell populations as inferred from CIBERTSORTx digital cytometry deconvolution, stratified by COVID-19 severity (severe vs. non-severe). We have made this clearer in the table title.

Reviewer #2

Using circulating protein profiling (of 48 select markers) and blood transcriptomics (of a subset of patients N=100), Cummings and colleagues assess the immune profiles of a COVID-19 adult cohort from Uganda (N=306). Patients span the disease severity spectrum from asymptomatic to severe disease with fatal outcomes. Three aspects of this study are of potential interest. First, the analysis of HIV+ vs. HIV- patients, although the numbers are on the low side and HIV status itself is not a strong correlate of disease severity, thus the signal that emerges are weak at best. Second, this study covers a patient cohort from sub-Saharan Africa, which is relatively understudied using immune profiling approaches. However, without comparative analysis with other populations, it is not clear whether any population specific immune response characteristics were uncovered here. For example, the disease severity correlates emerged are largely consistent with those reported previously in other populations. Third, the authors also conducted a comparative analysis of COVID-19 with other contemporaneous respiratory infection patients in the same region. The study design is a strength, however, the presence of many potential confounding variables including medications and timing (see also below) and the relatively small sample sizes limit the immunological insights that can be obtained. For example, ref. 2 should be an informative dataset to help assess some of the differences further in conjunction with the rich COVID-19 data sets available.

We thank the reviewer for their comments. With regards to our HIV comparisons, we acknowledge in the manuscript that the smaller sample sizes are a limitation. However, results generated using both soluble mediator and RNAseq data were generally consistent and provide key translational insights that warrant consideration in prior studies. We now frame this in the manuscript as follows (Lines 541-553):

“Three years into the COVID-19 pandemic, the immunological and clinical effects of HIV co-infection in COVID-19 remain incompletely understood. strengthening the interpretation of our findings. While key immunological features of severe COVID-19 (e.g., exaggerated innate immune activation, altered type 1 interferon responses, T-cell lymphopenia), may be exacerbated by HIV-related immune dysfunction, studies have shown comparable, and in some cases dampened, immune-inflammatory profiles in SARS-CoV-2/HIV co-infection [53-56]. In PLWH in our cohort, most of whom were virologically suppressed and had similar risk of severe illness, prognostic host responses were largely consistent when compared to those without HIV. Interestingly, however, in directly comparative analyses, PLWH showed a relatively diminished innate immune profile, including lower concentrations of IFN- α 2 and multiple pro-inflammatory mediators, with upregulation of cytotoxic T-cell activation. Despite these differences, higher concentrations of IFN- α 2, IL-6, and IL-1, all of which are associated with HIV-related immune activation, differentially scaled with more severe COVID-19 in PLWH.”

Unfortunately, we did not have access to biological samples from high-income countries to perform comparative analyses with this population. However, given profound differences in a multitude of demographic, host, viral, environmental, health system, and treatment variables between these settings, we feel that any biological comparisons would be imprecise.

For comparisons between patients with COVID-19 and influenza/non-influenza SARI, we adjusted our analyses by key demographic and clinical variables including age, sex, HIV status, and clinical severity. As corticosteroids were not used in the treatment of influenza/non-influenza SARI patients we did not include this exposure as a co-variable in our models. As we discuss in the manuscript, our results were consistent with those reported in high-income countries, with concentrations of IL-6 and other pro-inflammatory mediators significantly lower in COVID-19 compared to other severe respiratory infections (see Ref. 31). Moreover, our findings of unique associations between COVID-19 and high concentrations of pro-fibrotic growth factors reinforce laboratory studies suggesting that severe SARS-CoV-2 infection, as opposed to influenza, may reprogram macrophages towards profibrotic phenotypes (see Ref. 60). We believe that these observations reinforce the validity of our findings.

Regarding the helpful recommendation to increase discussion of similar, previously published reports, we address this comment in detail below.

Thus, while I applaud the authors’ efforts, a major limitation of this manuscript is the lack of conceptual and biological advances and insights, particularly given the extensive immune profiling data and observations already available on COVID-19 with 3+ years of intensive studies by many groups. The authors also did not cite many of the pioneering immune profiling papers, particularly those using and integrating circulating protein, multimodal single cell, transcriptomic, and metabolomic analyses that reported similar findings (e.g., see select references below – the authors should do a more thorough survey to ensure the key papers are cited).

We thank the reviewer for these comments. We have added a new paragraph to the discussion section comparing our main results with those of single-cell and genomic analyses of the COVID-19 host response with the relevant references added (Lines 458-476; see below). We have also emphasized these comparisons throughout multiple other sections of the discussion.

“Over the course of the pandemic, systems biology-based investigations have dissected the host response to SARS-CoV-2 with high resolution, revealing immunopathological signatures associated with COVID-19 severity and potential treatment targets. Consistent with our findings, multiple single-cell and genomic analyses of peripheral blood have identified dysregulated myeloid cell immunity as a key feature of severe COVID-19 [8,32,33]. In addition to increased circulation of activated, IFN-stimulated monocytes, emergency granulopoiesis and neutrophil-driven innate immune activation, often accompanied by expansion of aberrant, mature neutrophil subsets, has emerged as a common immunopathological finding in severe COVID-19 [8,32,33]. Our multimodal analyses reinforce a distinct role for neutrophil immunity in COVID-19 pathobiology, with neutrophil-driven pathways and neutrophil abundance highly enriched in severe COVID-19 and distinguishing this host response to that of influenza and other respiratory infections. Comprehensive single-cell analyses across the COVID-19 severity spectrum have also revealed increased quantities of circulating plasma cells/plasmablasts and IL-15-related exhaustion of circulating NK-cells in severe COVID-19, findings consistent with those observed in our cohort [33-35]. Reported T-cell signatures in severe COVID-19 have varied, with both extreme T-cell activation and T-cell lymphopenia, particularly of CD8+ cells, associated with severe illness and poor outcomes [10,36]. Our findings support this profile, with concurrent Th1-pathway upregulation and T-cell activation, along with CD8+ T-cell depletion, observed in patients with severe COVID-19.”

In addition, we believe that our multi-modal analyses across a long study period, spanning three SARS-CoV-2-variant driven phases of the pandemic, presents a major conceptual advance in understanding COVID-19 pathobiology. Although *in vitro* and computational experiments suggest that SARS-CoV-2 variants may differentially induce innate immune signaling and pro-inflammatory cytokine production, little remains known about these relationships *in vivo*, a knowledge gap which our study directly addressed.

Another concern is the cross-sectional nature of the analysis without longitudinal information. The systemic immune response to COVID-19 is highly dynamic across many variables. At the minimum, time since symptom onset should be included as a co-variate (e.g., the authors have also shown in fig 3a that some of the circulating proteins they examine are associated with time across patients.) Without longitudinal data it is not clear whether any of the group-based comparisons are confounded by response kinetics. For example, type I IFNs and downstream responses diminish quickly after peaking in blood early (line 178, see refs. 1 and 4 below). Similarly, while IL-15 levels can persist in the severe patients, IL-15 and associated transcriptional responses such as those in NK cells can be highly dynamic and dependent on disease severity (ref. 4 below). The same concerns apply to the comparison among COVID-19, influenza, and other respiratory infections (see ref. 2 below).

We agree that the cross-sectional nature of our analyses is a limitation of our study. We have made this clear in the limitations paragraph of the discussion (Lines 591-593). As suggested by the reviewer, we included symptom duration prior to enrollment as a covariable in key multivariable models and incorporated this variable into other unsupervised analyses. This includes our proportional odds models examining the association between immune mediator concentrations and COVID-19 clinical severity (Table S3, Lines 180-183) and logistic models examining the association between cluster-derived COVID-19 Response Signature assignment and Delta phase COVID-19 (Lines 394-397). Moreover, median duration of illness prior to

enrollment was identical among patients assigned to CRS-1 vs. CRS-2, suggesting that this variable is unlikely to drive differences between these subgroups (Table S13). Beyond these key analyses, we do not feel that adjusting all models for illness duration is optimal, as such an approach would require excluding asymptomatic patients (a clinically and biologically important group), thereby lowering sample sizes and power.

“Escalating severity” – this term was used in a few places in the manuscript. It implies temporal changes, but I think the authors meant to suggest associative changes between severe and less severe groups. Please clarify and be more specific.

We thank the reviewer for the opportunity to refine this terminology. We have removed the two uses of “escalating severity”, replacing this term with “increased severity,” and “more severe illness” (Lines 166, 171, 176-177).

A recurrent question I have throughout the manuscript is the extent by which the reported circulating protein marker or transcriptomic correlates are consistent with those reported by others. It is important to put findings into context, particularly in comparing to other populations/regions/countries including those from other regions of Africa.

We thank the reviewer for this comment. As above, we have added a new paragraph to the discussion section comparing our main results with those of single-cell and genomic analyses of the COVID-19 host response with the relevant references added (Lines 458-476). We have also emphasized these comparisons throughout multiple other sections of the discussion, including in the context of the only two other studies of similar nature from sub-Saharan Africa (Lines 541-546, Ref. 53-54).

How was the subset of 100 patients chosen for transcriptomic profiling?

As the pandemic evolved in real-time and Uganda experienced recurrent waves of COVID-19 (differentiated by varying levels of community transmission and epidemic peaks), we sought to collect whole-blood RNA, simultaneously with serum, from consecutively enrolled patients during each wave. As we ultimately stratified our study period based on SARS-CoV-2 variant-driven pandemic phases (using methodology consistent with that of U.S. CDC), we feel that the most appropriate description of the patients selected for transcriptomic profiling is a “subset of 100 who had whole-blood RNA samples collected simultaneously with serum during each phase of the pandemic.” We have now clarified this in the revised manuscript and have added a new table (Table S5) to the supplement presenting the characteristics of patients who underwent transcriptomic profiling. As per this table and as stated in the manuscript (Lines 209-211, demographics, HIV and Delta phase prevalence, and clinical severity in this group were comparable to the larger study population.

In some of the analyses, a comparison was made between severe patients and the rest. Why not use the entire disease severity spectrum?

We thank the reviewer for this comment and the opportunity to respond. Nearly all of our soluble immune mediator analyses are presented and analyzed using the entire severity spectrum (i.e., Kruskal Wallis-H testing and multivariable proportional odds models with clinical severity [asymptomatic, mild, moderate, severe] as the ordinal dependent variable. For our gradient-boosted machine models, we analyzed patients with severe vs. non-severe COVID-19 as these models, like other machine learning algorithms, have been optimized for binary classification.

Similarly, for our differential gene expression analyses, the most widely utilized platforms (i.e., DESeq2, used in our study) have been optimized for binary outcomes. Thus, given these methodological considerations, as well as the importance of gaining insights into biological features of patients with severe COVID-19 (the highest risk clinical state), we used binary comparisons of severe vs. non-severe where indicated. For analyses of immune mediators over the reported course of symptoms (Figure 3), patients with asymptomatic infection were excluded. Given the relatively low numbers of moderately ill patients and similarities in mediator profiles between moderate and mildly ill patients, we combined these groups to optimize comparisons with severely ill patients.

Line 215: “cell lines” – those are not cell lines.

We thank the reviewer for the opportunity to refine terminology. We have edited the text and now refer to these as “cell populations” (Line 217).

Line 218: not clear what “reducing thymic selection” meant here, particularly given that blood, not thymic tissues, was used. In general, the manuscript tends to list many GSEA based gene set labels that may or may not reflect the underlying complex biology being captured by cell states or circulating protein levels in blood. The authors should go over and interpret these carefully...

We thank the reviewer for this important comment. We agree that given concerns of the imprecise nature of the GSEA “Thymic T-cell selection” pathway, we have removed this pathway from Figure 2a and have edited the relevant text (Lines 218-221). However, we emphasize that we carefully reviewed all GSEA outputs and those presented in Figures 2a-2c, 4f, and 6b were included to highlight enrichment of pathologically important and diverse pathways, in accordance with similar high-quality manuscripts that have reported on pathway enrichment in COVID-19.

Line 240 and on: the broad downregulation of translational, ribosomal, and other biosynthetic processes has been reported in responses to viral infections including COVID-19. Anything specific to this population? Is this linked to the IFN related down-regulation of biosynthetic capacity to counter viral replication?

We thank the reviewer for this comment. We highlight in the manuscript (Lines 524-537) that in vitro experiments suggest that SARS-CoV-2 modulates host protein synthesis both to enhance viral mRNA translation and inhibit production of antiviral mediators. SARS-CoV-2 nonstructural proteins (e.g., Nsp1) are central to this process, accelerating degradation of host mRNAs and impairing nuclear mRNA export to attain a virally dominated mRNA pool. In parallel, Nsp1 preferentially inhibits host translation, including of type I interferons and other antiviral mediators, though blockade of mRNA entry channels and inhibitory binding to ribosomal subunits.

Overall, the cohort is quite male bias. It would be good to include a discussion on the caveats and implications of this bias.

We have added text to the discussion section of the manuscript (Lines 606-609) with a new reference, stating that “...most patients enrolled in our study were male. As differences in prognostically-relevant immune responses may be present between males and females with

COVID-19, further studies are needed to better define the role of sex in SARS-CoV-2 immunopathology [62].”

Reviewer #3

In this paper, the team provides a comprehensive plasma and whole blood transcriptomic analysis of a cohort of individuals with SARS-CoV-2 infection in Uganda. The report is well written and significantly adds to the literature with the clear characterization of the immunophenotype in this population which includes ~11% prevalence of concurrent HIV infection. The authors further add a comparison to individuals with influenza and other severe acute respiratory infections which adds specificity to their findings about COVID-19 pathogenesis. Overall, the paper adds key data on the pathogenesis of this infection outside of the high income country setting which is largely consistent with prior findings, but raises some questions for further exploration.

We thank the reviewer for these comments.

**A few additional details about the cohort would help interpretation of the findings:
-Methods description of how asymptomatic patients were identified – routine screening, testing at hospital presentation for other reasons etc.**

We have added additional details to the methods section of the manuscript (Lines 627-630) to clarify this point. Patients with SARS-CoV-2 infection nationwide were referred to our study site for management and monitored isolation, the latter for those with asymptomatic infection identified through routine screening at designated surveillance points (e.g., airports, border crossings).

-Timing from onset of symptoms to sample collection – these are explored in Table S2 and Figure 3, but should be included in the demographics table.

As suggested, we have added illness duration prior to enrollment to the main manuscript table (Table 1).

-Timing from use of corticosteroids to sampling (pre or post sampling)

Unfortunately, due to inconsistent documentation of medication administration timing, we do not have access to the precise timing of corticosteroid exposure. We have added this as a limitation to the limitations section of the manuscript discussion (Lines 607-609), stating that “...while we determined corticosteroid exposure for all patients, we were unable to precisely define the timing of administration relative to enrollment, which could have affected immune responses.”

-Separate table defining the characteristics of the subset of individuals included in the RNAseq experiments, currently text description suggests similar structure of distribution, but this should be presented in a table.

We have added a new table to the supplement (Table S5) presenting the characteristics of the subset of patients included in RNAseq analyses. As per this table and as stated in the manuscript (Lines 207-211), demographics, HIV and Delta phase prevalence, and clinical

severity in this group were comparable to the larger study population.

From a top level, it is not clear that the paper significantly addresses metabolism. The findings related to metabolic pathways are primarily around phagocytic respiratory burst and reactive oxygen species generation, which are likely related to neutrophil granule composition/degranulation. These are not per se related to the global regulation of immune cell metabolism as typically referenced to describe respiratory capacity in immune cell subsets. Would consider modifying that reference in the title.

We thank the author for this comment. We have replaced the term “immunometabolic” with “immune” in the title, which now reads: “Multimodal host profiling across the COVID-19 severity spectrum in Uganda reveals prognostic immune signatures that persist in HIV co-infection and diverge by variant-driven pandemic phase.” We have also replaced “immunometabolic” with “immune” throughout the manuscript.

Figure 1 - the models of proportional probabilities seem are limited by the very low numbers of individuals in the HIV positive group (severe with only 4) as suggested by the wide confidence intervals. The data presented in supplemental figure s2 are more clear, and would suggest either moving the subgroup severity analyses to supplemental or amending the text to indicate that these analyses have very limited power.

We thank the reviewer for this important comment. We have edited the text to emphasize the limited power in our proportional odds models analyzing the interaction between HIV status and immune mediators and risk of more severe COVID-19 (Lines 200-201).

Figure 3b – It appears that the reported p values here are the nominal p values and not the BH-adjusted ones in Supplemental Table s6. This should be clearly indicated as the other plots have presented adjusted p values. While this does mean that some of the reported comparisons are not significant in the fully adjusted model, I agree with the authors that the results are consistent with the overall cohort and the time course models in Figure 3a support this conclusion. But the lack of adjustment should be indicated.

We have made clear in the Figure 3b legend (Lines 1134-1135) that P-values in the figure reflect Wilcoxon rank-sum tests unadjusted for multiple comparisons with BH-adjusted P-values included in Table S7.

In the paragraph discussing the comparison of PLWH to the overall cohort, the text should be clarified to indicate that the majority of pathways in Figure 6b have FDR q values >0.1 and any associations are exploratory (currently written as “Beyond FDR q value” which is ambiguous).

We have edited the text to make clear that the majority of the pathways in Figure 6b have FDR q-values >0.10 (Line 415). We have also added text to the Figure 6 legend clarifying this (Lines 1217-1218).

The section on disruption of protein synthesis suggests that impaired transcription and cytosolic mitochondrial translation in patients with severe COVID-19 may be related directly to SARS-CoV-2 subversion of protein synthesis. The data cited to support this

are from human lung alveolar cell lines that are directly infected. It is unclear how this would relate to a global immune response profile in which the vast minority of cells are likely directly exposed to or infected by SARS-CoV-2. To suggest that the peripheral immune profile is in support of viral entry, the authors would need to demonstrate that this signature is linked to some measure of virus activity in the plasma or in these cells.page 10 lines 237-247. Would consider modifying this in the results and discussion or considering an alternate explanation of how infection and global inflammation may lead to a disruption in these pathways that is not directly dependent on infection of the cells with SARS-CoV-2.

We thank the reviewer for this important point. We have edited the corresponding section of the discussion substantially (Lines 524-537). The relevant text now states: “In vitro experiments in human alveolar epithelial cells suggest that SARS-CoV-2 modulates host protein synthesis, both to enhance viral mRNA translation and inhibit production of antiviral mediators [23]. SARS-CoV-2 nonstructural proteins (e.g., Nsp1) are central to this process, accelerating degradation of host mRNAs and impairing nuclear mRNA export to attain a virally dominated mRNA pool [23,50]. In parallel, Nsp1 preferentially inhibits host translation, including of type I interferons and other antiviral mediators, through blockade of mRNA entry channels and inhibitory binding to ribosomal subunits [51]. Consistent with these findings, we observed evidence of multifaceted impairment of cytoplasmic and mitochondrial protein synthesis in severe COVID-19. While nonstructural proteins of SARS-CoV-2 likely play a key role in infected alveolar epithelial cells, mechanisms underlying this observation in peripheral blood cells are unclear. Although SARS-CoV-2 can infect blood monocytes, we were unable to evaluate this in our samples. As has been reported in other severe viral infections, processes independent of viral cell entry, such as those activated in response to inflammatory or oxidative stress, may have blunted host translation [52].”

We have also edited text in the corresponding results section (Lines 237, 239) clarifying that the studies we refer to were in SARS-CoV-2-infected alveolar epithelial cells whereas our results were generated from peripheral blood samples.

Minor comments

References 15-18 which provide the data supporting high mortality rates in sub Saharan Africa are not cited in the text where they should be (line 111 which currently only has ref 13,14 listed). It is also of note that differences in mortality may reflect not a unique pattern of inflammation but rather resource constraints and this point is not clearly discussed.

We thank the reviewer for highlighting this. We have moved up these references (now 11-15) to this part of the manuscript (Lines 110-112). We have also added text to this sentence to acknowledge limited critical care capacity in the region. This sentence now states: “In sub-Saharan Africa, a low-income region where SARS-CoV-2 vaccine coverage remains poor and critical care capacity is limited, fatality rates for severe COVID-19 are among the highest in the world [11-15].”

The COVID-19 cohort is 75% male, which should be noted as a limitation, given the association of male sex and/or gender with risk of mortality. Although the models are

adjusted, minimal representation of females may limit the ability to identify relevant differences.

We have added text to the discussion section of the manuscript (Lines 606-609) with a new reference, stating that "...most patients enrolled in our study were male. As differences in prognostically-relevant immune responses may be present between males and females with COVID-19, further studies are needed to better define the role of sex in SARS-CoV-2 immunopathology [62]."

In the section on the NLRP3 inflammasome, a discussion of the specific leading edge genes in the inflammasome pathway that lead to this pathway enrichment would be of interest (Lines 320-333).

We have added the following text to the results section of the manuscript (Lines 325-330) to specify the 10 leading edge genes: "While activation of many of these processes was observed in patients with severe COVID-19, inflammasome assembly was highly enriched in patients with Delta phase COVID-19, with genes encoding multiple pattern recognition receptors (NLRP6, NLRP1, TLR6, TLR4, AIM2, TREM2), Pyrin (MEFV), caspase recruitment domain proteins (CARD8), cytoplasmic stress granules (DDX3X), and phospholipase C (PLCG2) comprising the core pathway enrichment set."

In Figure 5, panels g,i,k,l are somewhat redundant (i.e. oxygen use is part of the definition of severity) as they are all ways of demonstrating the enrichment of severe phenotypes in the CRS-2 group. Would consider choosing one way of presenting these data.

We thank the reviewer for this comment. Respectfully, we feel that each of these panels presents different indicators of clinical severity and outcomes and warrant inclusion in the figure. Panel 5g presents the proportional distribution of mild, moderate, and severe COVID-19 by CRS. Panel 5i presents the aggregate frequencies of not only severe COVID-19 or oxygen use but inability to ambulate (a widely utilized marker of severe illness and established predictor of inpatient mortality in resource-limited settings) and severely impaired functional status (as per Karnofsky Performance Status). Among patients who required oxygen therapy, Panel 5k presents the distribution of oxygen flow rates, showing that patients in CRS-2 required significantly higher levels of oxygen support, suggestive of more severe respiratory failure. Finally, panel 5l presents the cumulative incidence of poor in-hospital outcome among all patients with symptomatic COVID-19 and those only with severe illness.

REVIEWER COMMENTS

Reviewer #1 (Remarks to the Author):

The authors have well adapted their manuscript by correcting errors and addressing the concerns by extending the discussion.

Reviewer #3 (Remarks to the Author):

In this revision, the authors have sought to answer multiple of the issues raised in review. Although they have not been able to fully address all of the concerns, in part due to the limitations of available samples and data, they have substantively responded to the majority of the questions that were raised.

Specifically they have

- Included the time to symptom onset in Table 1 which shows the differences across severity
- Addressed the limitations of the lack of data on timing of corticosteroid administration in their discussion
- Added a table with the full characteristics of the subgroup with transcriptomics, which do appear comparable to the primary analytic group
- Amended the title to better represent the content of the paper (eliminating metabolic).

I do feel that the authors could still be more explicit in acknowledging the limitations of the numbers in the HIV analysis in both the reporting and the interpretation of those findings. However, they have added the appropriate statistical information to indicate where the results fall below corrected thresholds so that a careful reader can appreciate the statistical uncertainty.

Overall, while there are some parts of the paper that are overlapping with prior analyses (as raised by reviewer 2), there are unique aspects of the study, in particular related to the participants studied and the additional data from outside of resource rich settings.

Reviewer #4 (Remarks to the Author):

The study by Cummings et al. performed a system-level proteome and transcriptomics analysis of COVID-19 patients from Uganda. The research methodology employed in this study is commendable, representing one of its key strengths. Nevertheless, several factors could potentially muddle the results and restrict the conclusion of the study. I agree with reviewer#2 that factors encompassing a wide range of variables, such as medication usage and the timing of data collection, can potentially weaken the study. Furthermore, the study is hampered by its reliance on a relatively small sample size, specifically the PLWH. These limitations collectively impede how much we can draw meaningful insights, specifically towards the HIV interpretation. I have a few major points.

1. The transcriptomics were done on whole blood. As presented in the new Table S6, there were differences between the cell types Plasma, CD8, Monocytes, and Neutrophils between the severe and non-severe patients. For all the DeSeq analyses, these cell types should be adjusted.
2. There are many confounders in the small data set, and authors were adjusted for some. I suggest authors use the RUVseq (<https://bioconductor.org/packages/release/bioc/vignettes/RUVSeq/inst/doc/RUVSeq.html>) rather than conventional DeSeq2.
3. The immune mediator part, Table S4, shows that there are only four severe patients and six moderate, I don't think the statistics with those numbers give any insight. Moreover, HD48 has an Inter-assay variability ranging between 5-20% depending on the analyte and quality of samples.

Reviewer #1

The authors have well adapted their manuscript by correcting errors and addressing the concerns by extending the discussion.

Thank you.

Reviewer #3

In this revision, the authors have sought to answer multiple of the issues raised in review. Although they have not been able to fully address all of the concerns, in part due to the limitations of available samples and data, they have substantively responded to the majority of the questions that were raised.

Specifically they have

- Included the time to symptom onset in Table 1 which shows the differences across severity**
- Addressed the limitations of the lack of data on timing of corticosteroid administration in their discussion**
- Added a table with the full characteristics of the subgroup with transcriptomics, which do appear comparable to the primary analytic group**
- Amended the title to better represent the content of the paper (eliminating metabolic).**

I do feel that the authors could still be more explicit in acknowledging the limitations of the numbers in the HIV analysis in both the reporting and the interpretation of those findings. However, they have added the appropriate statistical information to indicate where the results fall below corrected thresholds so that a careful reader can appreciate the statistical uncertainty.

We thank the reviewer for their comments. We have added new text to the Results and Discussion sections of the manuscript emphasizing limitations in reporting and interpretation of the HIV analyses given the small number of persons living with HIV.

For the Results section (Lines 198-204), this text now states: "Although limited by the relatively small proportion of PLWH enrolled in our study, analyses using both methods were consistent with those in the larger cohort, suggesting that prognostic host responses in COVID-19 were generally conserved in PLWH (Table S4 and Figure S2 in supplement). While further limited by the small number of PLWH with severe illness, interaction models also suggested that higher concentrations of IFN- α 2, IL-6, CCL3, IL-1a, and IL-1Ra were associated with more severe COVID-19 in PLWH (Figures 1f and S3 in supplement)."

For the Discussion section (Lines 556-557), we have added a new sentence that states: "Given the relatively small proportion of PLWH in our cohort, these hypothesis-generating findings should be interpreted with caution."

Overall, while there are some parts of the paper that are overlapping with prior analyses (as raised by reviewer 2), there are unique aspects of the study, in particular related to the participants studied and the additional data from outside of resource rich settings.

Noted. We agree. Thank you.

Reviewer #4

The study by Cummings et al. performed a system-level proteome and transcriptomics analysis of COVID-19 patients from Uganda. The research methodology employed in this study is commendable, representing one of its key strengths. Nevertheless, several factors could potentially muddle the results and restrict the conclusion of the study. I agree with reviewer#2 that factors encompassing a wide range of variables, such as medication usage and the timing of data collection, can potentially weaken the study. Furthermore, the study is hampered by its reliance on a relatively small sample size, specifically the PLWH. These limitations collectively impede how much we can draw meaningful insights, specifically towards the HIV interpretation. I have a few major points.

We thank the reviewer for these comments. First, regarding the timing of sample collection, we included symptom duration prior to enrollment (i.e., sample collection) as a covariable in key multivariable models and incorporated this variable into other unsupervised analyses. This includes our proportional odds models examining the association between immune mediator concentrations and COVID-19 clinical severity (Table S3) and logistic models examining the association between cluster-derived COVID-19 Response Signature assignment and Delta phase COVID-19 (Lines 397-400). Moreover, median duration of illness prior to enrollment was identical among patients assigned to CRS-1 vs. CRS-2, suggesting that this variable is unlikely to drive differences between these subgroups (Table S13).

Second, while we determined corticosteroid exposure for all patients, we do not have access to the precise timing of corticosteroid exposure due to inconsistent documentation of medication administration timing. This limitation is included in the corresponding section of the manuscript discussion (Lines 613-615), stating that "...while we determined corticosteroid exposure for all patients, we were unable to precisely define the timing of administration relative to enrollment, which could have affected immune responses."

In response to comments from Reviewer #3, we have added new text to the Results and Discussion sections of the manuscript emphasizing limitations in reporting and interpretation of the HIV analyses given the small number of persons living with HIV.

For the Results section (Lines 198-204), this text now states: "Although limited by the relatively small proportion of PLWH enrolled in our study, analyses using both methods were consistent with those in the larger cohort, suggesting that prognostic host responses in COVID-19 were generally conserved in PLWH (Table S4 and Figure S2 in supplement). While further limited by the small number of PLWH with severe illness, interaction models also suggested that higher concentrations of IFN- α 2, IL-6, CCL3, IL-1a, and IL-1Ra were associated with more severe COVID-19 in PLWH (Figures 1f and S3 in supplement)."

For the Discussion section (Lines 556-557), we have added a new sentence that states: "Given the relatively small proportion of PLWH in our cohort, these hypothesis-generating findings should be interpreted with caution."

As discussed below, we have also further adjusted our primary immune mediator and gene expression analysis (severe vs. non-severe COVID-19) by SARS-CoV-2 phase (updated Figures 2A-2C and Tables S2-S3). These analyses are now adjusted for age, sex, HIV status,

SARS-CoV-2 phase, and pre-enrollment illness duration (the latter in Table S3). The results are consistent with our prior findings.

1. The transcriptomics were done on whole blood. As presented in the new Table S6, there were differences between the cell types Plasma, CD8, Monocytes, and Neutrophils between the severe and non-severe patients. For all the DeSeq analyses, these cell types should be adjusted.

We thank the reviewer for this comment. Unfortunately, our study was not powered for cell-type specific differential gene expression analyses. Simulation studies show that the two most commonly used algorithms designed for cell type-specific gene expression analysis of RNAseq data (CARseq, TOAST) are under-powered at our sample size (Jin et al., Nat Comput Sci. 2021, PMID: 34957416). TOAST does not provide fold changes, which would also prevent us from applying pathway enrichment analysis. Even for CARseq, the more powerful algorithm, there would be considerable uncertainty in the estimates for cell-type specific gene expression in our study population. In contrast to CARseq and TOAST, DESeq2 was not developed to incorporate cell type proportion data inferred from digital cytometry deconvolution. Further, recent evidence suggests that use of DESeq2 for cell type-specific differential gene expression results in poorly accurate between-group estimates (Jaakkola et al., Brief Bioinform. 2022, PMID: 34651640). Thus, we feel it is most appropriate to present our bulk DESeq2 and pathway analyses alongside CIBERSORTx-inferred immune cell populations. This widely accepted approach has been used to dissect the host response in a variety of infectious and respiratory diseases, including COVID-19 (e.g., Wang et al., Nat Commun, 2022, PMID: 35995775; Kariotis et al., Nat Commun. 2021, PMID: 34876579; Hanley et al., Nat Commun, 2021, PMID: 34031380). Nonetheless, we have emphasized this as a limitation of our analyses in the discussion section of the updated manuscript (Lines 602-604).

2. There are many confounders in the small data set, and authors were adjusted for some. I suggest authors use the RUVseq (<https://bioconductor.org/packages/release/bioc/vignettes/RUVSeq/inst/doc/RUVSeq.html>) rather than conventional DeSeq2.

We thank the reviewer for this comment. Regarding use of DESeq2 vs. RUVseq, we respectfully feel that use of the former is most appropriate. While RUVseq can be useful if the covariables for adjustment are unknown (e.g., hidden batch effects), use of DESeq2 is appropriate if relevant covariables are understood based on subject matter knowledge. Further, if RUVseq were to be applied to our RNAseq data, “in-silico empirical” negative controls (e.g., genes least significantly differentially expressed between conditions) would have to be used. As these genes are not truly negative controls (since there is, by definition, differential expression across conditions), such an approach is likely to remove actual biological signals from the analyses.

Thus, following recommendations for multivariable modeling and to optimize the interpretability of our findings, we included co-variables (i.e., “unwanted variation factors”) in adjusted DESeq2 models based on relevance to host responses and outcomes in COVID-19 (e.g., age, sex, HIV status, illness duration prior to enrollment/sample collection, etc.). This is a widely accepted approach. In the updated manuscript, we have further adjusted our primary DESeq2 comparison (severe vs. non-severe COVID-19) by SARS-CoV-2 phase (updated Figures 2A-2C) to ensure that our primary results are consistent across study periods. This model is now

adjusted for age, sex, HIV status, and SARS-CoV-2 phase. The results are consistent with our prior findings.

3. The immune mediator part, Table S4, shows that there are only four severe patients and six moderate, I don't think the statistics with those numbers give any insight. Moreover, HD48 has an Inter-assay variability ranging between 5-20% depending on the analyte and quality of samples.

We have added new text to the Results and Discussion sections of the manuscript emphasizing limitations in reporting and interpretation of the HIV analyses given the small number of persons living with HIV.

For the Results section (Lines 198-204), this text now states: "Although limited by the relatively small proportion of PLWH enrolled in our study, analyses using both methods were consistent with those in the larger cohort, suggesting that prognostic host responses in COVID-19 were generally conserved in PLWH (Table S4 and Figure S2 in supplement). While further limited by the small number of PLWH with severe illness, interaction models also suggested that higher concentrations of IFN- α 2, IL-6, CCL3, IL-1a, and IL-1Ra were associated with more severe COVID-19 in PLWH (Figures 1f and S3 in supplement)."

For the Discussion section (Lines 556-557), we have added a new sentence that states: "Given the relatively small proportion of PLWH in our cohort, these hypothesis-generating findings should be interpreted with caution."

We note the reviewer's comment about the inter-assay variability of the Human Cytokine/Chemokine Panel A 48-Plex Discovery Assay Array (HD48). This and related assays have been employed in thousands of translational studies investigating human immune responses. Moreover, as discussed in the Methods section of the manuscript (Line 704), mediator concentrations were quantified in duplicate with the mean used for analysis.

REVIEWER COMMENTS

Reviewer #3 (Remarks to the Author):

In this second resubmission, the authors have more completely addressed the concerns around the limitations of the sample of individuals with HIV in their study set. They have also updated their analyses to adjust for multiple confounders in Figure 2. Review of that figure indicates minor changes in the ordering of pathway enrichment from the prior version, but largely consistent findings as they indicate in the text.

They have substantially addressed this remaining critique of their work which remains of significant value.

Reviewer #4 (Remarks to the Author):

Cummings et al. submitted the response, but I find it somewhat lacking in addressing my concerns adequately, similar to how it failed to fully tackle the issues raised in the previous review by Rev#2. It seems that the response could benefit from a more comprehensive approach that directly engages with the core points of contention, addressing them in a more thorough and conclusive manner. By delving deeper into the specific concerns highlighted, can effectively clarify any ambiguities or uncertainties surrounding the findings.

1. Despite the utilization of CIBERSORT, it appears that the authors are hinting that there is no power, then why to perform the analysis.

2. It's worth noting that RUVseq incorporates the functionality of DESeq2, streamlining the initial preprocessing phase by normalizing hidden factors. At the end RUVseq will use DeSeq2 only for differential gene expression.

3. I maintain reservations about the validity of the statistical conclusions drawn from the relatively small (4 and 6 in each group) sample size (though authors claims it's hypothesis generation). Even to generate hypothesis there should be some numbers, not just 4 and 6 samples in each experimental groups.

Reviewer #3

In this second resubmission, the authors have more completely addressed the concerns around the limitations of the sample of individuals with HIV in their study set. They have also updated their analyses to adjust for multiple confounders in Figure 2. Review of that figure indicates minor changes in the ordering of pathway enrichment from the prior version, but largely consistent findings as they indicate in the text.

Noted, thank you.

They have substantially addressed this remaining critique of their work which remains of significant value.

Noted, thank you.

Reviewer #4

Cummings et al. submitted the response, but I find it somewhat lacking in addressing my concerns adequately, similar to how it failed to fully tackle the issues raised in the previous review by Rev#2. It seems that the response could benefit from a more comprehensive approach that directly engages with the core points of contention, addressing them in a more thorough and conclusive manner. By delving deeper into the specific concerns highlighted, can effectively clarifies any ambiguities or uncertainties surrounding the findings.

We thank the reviewer for their comments. We have sought to address the questions regarding bioinformatics methods more fully below.

1. Despite the utilization of CIBERSORT, it appears that the authors are hinting that there is no power, then why to perform the analysis.

We appreciate the opportunity to clarify this point, specifically the distinction between cell type-specific differential gene expression (DGE) analysis and between-group comparisons of immune cell abundances as inferred from digital cytometry deconvolution.

To recap our approach, we performed DGE and pathway enrichment analysis on our bulk RNAseq data (e.g., between patients with severe vs. non-severe COVID-19). Separately, we inferred the abundance of key immune cell types by applying the CIBERSORTx digital cytometry deconvolution algorithm to our bulk RNAseq data. We then compared the abundance of these cell types between different patient groups (e.g., those with severe vs. non-severe COVID-19). As highlighted in our prior Reviewer Responses, performing and reporting these analyses separately (DGE/pathway enrichment and inferred immune cell type comparisons), as we have done, is a widely accepted approach, including in recently published analyses of infectious and respiratory diseases such as COVID-19 (e.g., Wang et al., Nat Commun, 2022, PMID: 35995775; Kariotis et al., Nat Commun. 2021, PMID: 34876579; Hanley et al., Nat Commun, 2021, PMID: 34031380).

In contrast, cell type-specific DGE analysis is a distinct method that directly incorporates immune cell type proportions (such as those inferred by CIBERSORTx or other digital cytometry pipelines) into DGE algorithms. As highlighted in the prior Reviewer Responses, simulation studies show that the two most common cell type-specific DGE algorithms (CARseq, TOAST)

are under-powered at our sample size (Jin et al., Nat Comput Sci. 2021, PMID: 34957416, Jaakkola et al., Brief Bioinform. 2022, PMID: 34651640).

Thus, we feel it is most appropriate to present our bulk DGE and pathway enrichment analyses separately from our CIBERSORTx-inferred immune cell population comparisons, in line with similar studies mentioned above. Further, throughout our analyses, significant differences in immune cell types between patient groups (e.g., severe vs. non-severe COVID-19, Delta vs. non-Delta COVID-19) mirror findings from our DGE/pathway and soluble immune mediator analyses, suggesting that they are adequately powered and biologically consistent.

2. It's worth noting that RUVseq incorporates the functionality of DESeq2, streamlining the initial preprocessing phase by normalizing hidden factors. At the end RUVseq will use DESeq2 only for differential gene expression.

We appreciate the Reviewer's comments regarding RUVseq and the compatibility of this algorithm with DESeq2. Respectfully, however, we do not feel that preprocessing with RUVseq is appropriate for our data. We maintain that our current DESeq2-based normalization and DGE methods allow us to both mitigate between-sample variation introduced by possible technical effects and identify important biological signals in our data. First, as we highlighted in our prior Reviewer Responses, RUVseq can be useful to adjust for technical batch effects that may not be obvious, particularly when data are pooled across different laboratories or sequencing platforms (a circumstance not applicable to our study). However, the attached relative log expression plot of our DESeq2-normalized data does not suggest the presence of considerable residual technical/batch effects (i.e., the boxplots are centered on zero and generally of comparable size). Second, use of RUVseq requires a set of negative control genes. If RUVseq were to be applied to our dataset, "in-silico empirical" negative controls (i.e., genes least significantly differentially expressed between conditions) would have to be used. As these genes are not truly negative controls (since there is, by definition, differential expression across conditions) and the significance threshold for their categorization is variable, such an approach may remove actual biological signals from the analyses. As we have encountered this in prior studies, we do not feel it is an optimal method for our current study and strongly favor our current approach.

3. I maintain reservations about the validity of the statistical conclusions drawn from the relatively small (4 and 6 in each group) sample size (though authors claims it's hypothesis generation). Even to generate hypothesis there should be some numbers, not just 4 and 6 samples in each experimental groups.

We acknowledge the small numbers of patients included in the HIV-stratified analysis. As part of Revision #2, we included additional text in the manuscript to emphasize this as a limitation and encouraged readers to view these findings, which we maintain are reasonably hypothesis-generating, with caution.

These statements can be found in the current manuscript as follows:

For the Results section, the relevant text states: “Although limited by the relatively small proportion of PLWH enrolled in our study, analyses using both methods were consistent with those in the larger cohort, suggesting that prognostic host responses in COVID-19 were generally conserved in PLWH (Table S4 and Figure S2 in supplement). While further limited by the small number of PLWH with severe illness, interaction models also suggested that higher concentrations of IFN- α 2, IL-6, CCL3, IL-1a, and IL-1Ra were associated with more severe COVID-19 in PLWH (Figures 1f and S3 in supplement).”

For the Discussion section, the relevant text states: “Given the relatively small proportion of PLWH in our cohort, these hypothesis-generating findings should be interpreted with caution.”